# Stable Nonconvex-Nonconcave Training
# via Linear Interpolation

**Thomas Pethick**
EPFL (LIONS)
thomas.pethick@epfl.ch

**Wanyun Xie**
EPFL (LIONS)
wanyun.xie@epfl.ch

**Volkan Cevher**
EPFL (LIONS)
volkan.cevher@epfl.ch

## Abstract

This paper presents a theoretical analysis of linear interpolation as a principled method for stabilizing (large-scale) neural network training. We argue that instabilities in the optimization process are often caused by the nonmonotonicity of the loss landscape and show how linear interpolation can help by leveraging the theory of nonexpansive operators. We construct a new optimization scheme called relaxed approximate proximal point (RAPP), which is the first 1-SCLI method to achieve last iterate convergence rates for $\rho$-comonotone problems while only requiring $\rho > -\frac{1}{2L}$. The construction extends to constrained and regularized settings. By replacing the inner optimizer in RAPP we rediscover the family of Lookahead algorithms for which we establish convergence in cohypomonotone problems even when the base optimizer is taken to be gradient descent ascent. The range of cohypomonotone problems in which Lookahead converges is further expanded by exploiting that Lookahead inherits the properties of the base optimizer. We corroborate the results with experiments on generative adversarial networks which demonstrates the benefits of the linear interpolation present in both RAPP and Lookahead.

## 1 Introduction

Stability is a major concern when training large scale models. In particular, generative adversarial networks (GANs) are known to be notoriously difficult to train. To stabilize training, the Lookahead algorithm of Zhang et al. (2019) was recently proposed for GANs Chavdarova et al. (2020) which linearly interpolates with a slow moving iterate. The mechanism has enjoyed superior empirical performance in both minimization and minimax problems, but it largely remains a heuristic with little theoretical motivation.

One major obstacle for providing a theoretical treatment, is in capturing the (fuzzy) notion of stability. Loosely speaking, a training dynamics is referred to as *unstable* in practice when the iterates either cycle indefinitely or (eventually) diverge—as has been observed for the Adam optimizer (see e.g. Gidel et al. (2018, Fig. 12) and Chavdarova et al. (2020, Fig. 6) respectively). Conversely, a *stable* dynamics has some bias towards stationary points. The notion of stability so far (e.g. in Chavdarova et al. (2020, Thm. 2-3)) is based on the spectral radius and thus inherently *local*.

In this work, we are interested in establishing *global* convergence properties, in which case some structural assumptions are needed. One (nonmonotone) structure that lends itself well to the study of stability is that of cohypomonotonicity studied in Combettes & Pennanen (2004); Diakonikolas et al. (2021), since even the extragradient method has been shown to cycle and diverge in this problem class (see Pethick et al. (2022, Fig. 1) and Pethick et al. (2023, Fig. 2) respectively). We provide a geometric intuition behind these difficulties in Figure 1. Biasing the optimization schemes towards stationary points becomes a central concern and we demonstrate in Figure 2 that Lookahead can indeed converge for such nonmonotone problems.

37th Conference on Neural Information Processing Systems (NeurIPS 2023).

Table 1: Overview of last iterate results with our contribution highlighted in blue . Prior to this work there existed no rates for 1-SCLI schemes handling $\rho$-comonotone problems with $\rho \in (-1/2L, \infty)$ and no global convergence guarantees for Lookahead beyond bilinear games.

| | Method | Setting | $\rho$ | Handles constraints | $\rho$-independent rates | Reference |
|---|---|---|---|---|---|---|
| Implicit | PP | Comonotone | $(-1/2L, \infty)$ | ✓ | ✗ | (Gorbunov et al., 2022b, Thm. 3.1) |
| | Relaxed PP | Comonotone | $(-1/2L, \infty)$ | ✓ | ✓ | Theorem 6.2 |
| Extrapolate | EG | Comonotone & Lips. | $(-1/8L, \infty)$ | ✗ | ✗ | (Gorbunov et al., 2022b, Thm. 4.1) |
| | EG+ | Comonotone & Lips. | | | Unknown rates | |
| | RAPP | Comonotone & Lips. | $(-1/2L, \infty)$ | ✓ | ✓ | Corollary 6.4 |
| Lookahead | LA-GDA | Local | - | ✗ | - | (Chavdarova et al., 2020, Thm. 2) |
| | | Bilinear | - | ✗ | - | (Ha & Kim, 2022, Cor. 7) |
| | | Comonotone & Lips. | $(-1/3\sqrt{3}L, \infty)$ | ✗ | - | Theorem 7.1 |
| | LA-EG | Bilinear | - | ✗ | - | (Ha & Kim, 2022, Cor. 8) |
| | | Monotone & Lips. | - | ✓ | - | Theorem F.1 |
| | LA-CEG+ | Comonotone & Lips. | $(-1/2L, \infty)$ | ✓ | - | Corollary 7.7 |

A principled approach to cohypomonotone problems is the extragradient+ algorithm (EG+) proposed by Diakonikolas et al. (2021). However, the only known rates are on the best iterate, which can be problematic to pick in practice. It is unclear whether *last* iterate rates for EG+ are possible even in the monotone case (see discussion prior to Thm. 3.3 in Gorbunov et al. (2022a)). For this reason, the community has instead resorted to showing last iterate of extragradient (EG) method of Korpelevich (1977), despite originally being developed for the monotone case. Maybe not surprisingly, EG only enjoys a last iterate guarantee under mild form of cohypomonotonicity and have so far only been studied in the unconstrained case (Luo & Tran-Dinh; Gorbunov et al., 2022b). Recently, last iterate rate were established for the same (tight) range of cohypomonotone problems for which EG+ has best iterate guarantees. However, the analyzed scheme is *implicit* and the complexity blows up with increasing cohypomonotonicity (Gorbunov et al., 2022b). This leaves the questions: *Can an explicit scheme enjoy last iterate rates for the same range of cohypomonotone problems? Can the rate be agnostic to the degree of cohypomonotonicity?* We answer both in the affirmative.

This work focuses on 1-SCLI schemes (Arjevani et al., 2015; Golowich et al., 2020), whose update rule only depends on the previous iterate in a time-invariant fashion. Another approach to establishing last iterate is Halpern-type methods with an explicit scheme developed in Lee & Kim (2021) for cohypomonotone problems and later extended to the constrained case in Cai et al. (2022) (c.f. Appendix A).

As will become clear, a principled mechanism behind convergence in this nonmonotone class is the linear interpolation also used in Lookahead. This iterative interpolation is more broadly referred to as the Krasnosel'skiĭ-Mann (KM) iteration in the theory of nonexpansive operators. We show that the extragradient+ algorithm (EG+) of Diakonikolas et al. (2021), our proposed relaxed approximate proximal point method (RAPP), and Lookahead based algorithms are all instances of the (inexact) KM iteration and provide simple proofs of these schemes in the cohypomonotone case.

More concretely we make the following contributions:

1. We prove global convergence rates for the last iterate of our proposed algorithm RAPP which additionally handles constrained and regularized settings. This makes RAPP the first 1-SCLI scheme to have non-asymptotic guarantees for $\rho$-comonotone problems while only requiring $\rho > -1/2L$. As a byproduct we obtain a last iterate convergence rate for an implicit scheme that is *independent* of the degree of cohypomonotonicity. The last iterate rates are established by showing monotonic decrease of the operator norm–something which is not possible for EG+. This contrast is maybe surprising, since RAPP can be viewed as an extension of EG+, which simply takes multiple extrapolation steps.

2. By replacing the inner optimization routine in RAPP with gradient descent ascent (GDA) and extragradient (EG) we rediscover the Lookahead algorithms considered in Chavdarova et al. (2020). We obtain guarantees for the Lookahead variants by deriving nonexpansive properties of the base optimizers. By casting Lookahead as a KM iteration we find that the optimal interpolation constant is $\lambda = 0.5$. This choice corresponds to the default value used in practice for both minimization and minimax—thus providing theoretical motivation for the parameter value.

3. For $\tau = 2$ inner iterations we observe that LA-GDA reduces to a linear interpolation between GDA and EG+ which allows us to obtain global convergence in $\rho$-comonotone problems when $\rho > -1/3\sqrt{3}L$. However, for $\tau$ large, we provide a counterexample showing that LA-GDA cannot be guaranteed to converge. This leads us to instead propose LA-CEG+ which corrects the inner optimization to guarantee global convergence for $\rho$-comonotone problems when $\rho > -1/2L$.

4. We test the methods on a suite of synthetic examples and GAN training where we confirm the stabilizing effect. Interestingly, RAPP seems to provide a similar benefit as Lookahead, which suggest that linear interpolation could play a key role also experimentally.

An overview of the theoretical results is provided in Table 1 and Figure 5§B.

## 2 Related work

**Lookahead**   The Lookahead algorithm was first introduced for minimization in Zhang et al. (2019). In the context of Federated Averaging in federated learning (McMahan et al., 2017) and the Reptile algorithm in meta-learning (Nichol et al., 2018), the method can be seen as a single worker and single task instance respectively. Analysis for Lookahead was carried out for nonconvex minimization (Wang et al., 2020; Zhou et al., 2021) and a nested variant proposed in (Pushkin & Barba, 2021). Chavdarova et al. (2020) popularized the Lookahead algorithm for minimax training by showing state-of-the-art performance on image generation tasks. Apart from the original local convergence analysis in Chavdarova et al. (2020) and the bilinear case treated in Ha & Kim (2022) we are not aware of any convergence analysis for Lookahead for minimax problems and beyond.

**Cohypomonotone**   Cohypomontone problems were first studied in Iusem et al. (2003); Combettes & Pennanen (2004) for proximal point methods and later expanded on in greater detail in Bauschke et al. (2021). The condition was relaxed to the star-variant referred to as the weak Minty variational inequality (MVI) in Diakonikolas et al. (2021) and the extragradient+ algorithm (EG+) was analyzed. The analysis of EG+ was later tightened and extended to the constrained case in Pethick et al. (2022).

**Proximal point**   The proximal point method (PP) has a long history. For maximally monotone operators (and thus convex-concave minimax problems) convergence of PP follows from Opial (1967). The first convergence analysis of *inexact* PP dates back to Rockafellar (1976); Brézis & Lions (1978). It was later shown that convergence also holds for the *relaxed* inexact PP as defined in (8) (Eckstein & Bertsekas, 1992). In recent times, PP has gained renewed interest due to its success for certain nonmonotone structures. Inexact PP was studied for cohypomontone problems in Iusem et al. (2003). Asymptotic convergence was established of the relaxed inexact PP for a sum of cohypomonotone operators in Combettes & Pennanen (2004), and later considered in Grimmer et al. (2022) without inexactness. Last iterate rates were established for PP in $\rho$-comonotone problems (with a dependency on $\rho$) (Gorbunov et al., 2022b). Explicit approximations of PP through a contractive map was used for convex-concave minimax problems in Cevher et al. (2023) and was the original motivation for MirrorProx of Nemirovski (2004). See Appendix A for additional references in the stochastic setting.

## 3 Setup

We are interested in finding a zero of an operator $S : \mathbb{R}^d \rightrightarrows \mathbb{R}^d$ which decomposes into a Lipschitz continuous (but possibly nonmonotone) operator $F$ and a maximally monotone operator $A$, i.e. find $z \in \mathbb{R}^d$ such that,

$$0 \in Sz := Az + Fz. \qquad (1)$$

Most relevant in the context of GAN training is that (1) includes constrained minimax problems.

**Example 3.1.** *Consider the following minimax problem*

$$\min_{x \in \mathcal{X}} \max_{y \in \mathcal{Y}} \phi(x, y). \qquad (2)$$

*The problem can be recast as the inclusion problem* (1) *by defining the joint iterates $z = (x, y)$, the stacked*

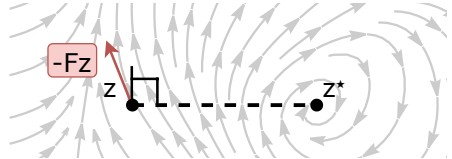

Figure 1: Consider $\min_{x \in \mathcal{X}} \max_{y \in \mathcal{Y}} \phi(z)$ with $z = (x, y)$. As opposed to convex-concave minimax problems, the cohypomonotone condition allows the gradients $Fz = (\nabla_x \phi(z), -\nabla_y \phi(z))$ to point away from the solutions (see Appendix B.1 for the relationship between cohypomonotonicity and the weak MVI). This can lead to instability issues for standard algorithms such as the Adam optimizer.

*gradients* $Fz = (\nabla_x \phi(x, y), -\nabla_y \phi(x, y))$, *and* $A = (\mathcal{N}_\mathcal{X}, \mathcal{N}_\mathcal{Y})$ *where* $\mathcal{N}$ *denotes the normal cone. As will become clear (cf. Algorithm 1), $A$ will only be accessed through the resolvent* $J_{\gamma A} := (\mathrm{id} + \gamma A)^{-1}$ *which reduces to the proximal operator. More specifically* $J_{\gamma A}(z) = (\mathrm{proj}_\mathcal{X}(x), \mathrm{proj}_\mathcal{Y}(y))$.

We will rely on the following assumptions (see Appendix B for any missing definitions).

**Assumption 3.2.** *In problem* (1),

(i) *The operator* $A : \mathbb{R}^d \rightrightarrows \mathbb{R}^d$ *is maximally monotone.*

(ii) *The operator* $F : \mathbb{R}^d \to \mathbb{R}^d$ *is $L$-Lipschitz, i.e. for some* $L \in [0, \infty)$,

$$\|Fz - Fz'\| \le L\|z - z'\| \quad \forall z, z' \in \mathbb{R}^d.$$

(iii) *The operator* $S := F + A$ *is $\rho$-comonotone for some* $\rho \in (-1/2L, \infty)$, *i.e.*

$$\langle v - v', z - z' \rangle \ge \rho\|v - v'\|^2 \quad \forall(v, z), (v', z') \in \mathrm{grph}\, S.$$

*Remark* 3.3. Assumption 3.2(iii) is also known as $|\rho|$-cohypomonotonicity when $\rho < 0$, which allows for increasing nonmonotonicity as $|\rho|$ grows. See Appendix B.1 for the relationship with weak MVI.

When only stochastic feedback $\hat{F}_\sigma(\cdot, \xi)$ is available we make the following classical assumptions.

**Assumption 3.4.** *For the operator* $\hat{F}_\sigma(\cdot, \xi) : \mathbb{R}^d \to \mathbb{R}^d$ *the following holds.*

(i) *Unbiased:* $\mathbb{E}_\xi[\hat{F}_\sigma(z, \xi)] = Fz \quad \forall z \in \mathbb{R}^d.$

(ii) *Bounded variance:* $\mathbb{E}_\xi[\|\hat{F}_\sigma(z, \xi) - Fz\|^2] \le \sigma^2 \quad \forall z, z' \in \mathbb{R}^d.$

# 4 Inexact Krasnosel'skiĭ-Mann iterations

The main work horse we will rely on is the inexact Krasnosel'skiĭ-Mann (IKM) iteration from monotone operators (also known as the *averaged* iteration), which acts on an operator $T : \mathbb{R}^d \to \mathbb{R}^d$ with inexact feedback,

$$z^{k+1} = (1 - \lambda)z^k + \lambda(Tz^k + e^k), \tag{IKM}$$

where $\lambda \in (0, 1)$ and $e^k$ is a random variable with dependency on all variables up until (and including) $k$. The operator $\widetilde{T}_k : z \mapsto Tz + e^k$ can crucially be an iterative optimization scheme in itself. This is important, since we can obtain RAPP, LA-GDA and LA-CEG+ by plugging in different optimization routines. In fact, RAPP is derived by taking $\widetilde{T}_k$ to be a (contractive) fixed point iteration in itself, which approximates the resolvent.

We note that also the extragradient+ (EG+) method of Diakonikolas et al. (2021), which converges for cohypomonotone and Lipschitz problems, can be seen as a Krasnosel'skiĭ-Mann iteration on an extragradient step

$$\begin{aligned} \mathrm{EG}(z) &= z - \gamma F(z - \gamma Fz) \\ z^{k+1} &= (1 - \lambda)z^k + \lambda\,\mathrm{EG}(z^k) \end{aligned} \tag{EG+}$$

where $\lambda \in (0, 1)$. We provide a proof of EG+ in Theorem G.1 which extends to the constrained case using the construction from Pethick et al. (2022) but through a simpler argument under fixed stepsize.

Essentially, the IKM iteration leads to a conservative update that stabilizes the update using the previous iterate. This is the key mechanism behind showing convergence in the nonmonotone setting known as cohypomonotonicity. Very generally, it is possible to provide convergence guarantees for IKM when the following holds (Theorem C.1 is deferred to the appendix due to space limitations).

**Definition 4.1.** *An operator* $T : \mathbb{R}^n \to \mathbb{R}^d$ *is said to be quasi-nonexpansive if*

$$\|Tz - z'\| \le \|z - z'\| \quad \forall z \in \mathbb{R}^d, \forall z' \in \mathrm{fix}\, T. \tag{3}$$

*Remark* 4.2. This notion is crucial to us since the resolvent $J_B := (\mathrm{id} + B)^{-1}$ is (quasi)-nonexpansive if $B$ is $1/2$-cohypomonotone (Bauschke et al., 2021, Prop. 3.9(iii)).

# 5 Approximating the resolvent

As apparent from Remark 4.2, the IKM iteration would provide convergence to a zero of the cohypomonotone operator $S$ from Assumption 3.2 by using its resolvent $T = J_{\gamma S}$. However, the update is implicit, so we will instead approximate $J_{\gamma S}$. Given $z \in \mathbb{R}^d$ we seek $z' \in \mathbb{R}^d$ such that

$$z' = J_{\gamma S}(z) = (\mathrm{id} + \gamma S)^{-1} z = (\mathrm{id} + \gamma A)^{-1}(z - \gamma F z')$$

This can be approximated with a fixed point iteration of

$$Q_z : w \mapsto (\mathrm{id} + \gamma A)^{-1}(z - \gamma F w) \tag{4}$$

which is a contraction for small enough $\gamma$ since $F$ is Lipschitz continuous. It follows from Banach's fixed-point theorem Banach (1922) that the sequence converges linearly. We formalize this in the following theorem, which additionally applies when only stochastic feedback is available.

$$w^{t+1} = (\mathrm{id} + \gamma A)^{-1}(z - \gamma \hat{F}_\sigma(w^t, \xi_t)) \quad \xi_t \sim \mathcal{P} \tag{5}$$

**Lemma 5.1.** *Suppose Assumptions 3.2(i), 3.2(ii) and 3.4. Given $z \in \mathbb{R}^d$, the iterates generated by* (5) *with $\gamma \in (0, 1/L)$ converges to a neighborhood linearly, i.e.,*

$$\mathbb{E}\left[\|w^\tau - J_{\gamma S}(z)\|^2\right] \leq (\gamma L)^{2\tau}\|w^0 - w^\star\|^2 + \frac{\gamma^2}{(1-\gamma L)^2}\sigma^2. \tag{6}$$

The resulting update in (5) is identical to GDA but crucially always steps from $z$. We use this as a subroutine in RAPP to get convergence under a cohypomonotone operator while only suffering a logarithmic factor in the rate.

**Interpretation** In the special case of the constrained minimax problem in (2), the application of the resolvent $J_{\gamma S}(z)$ is equivalent to solving the following optimization problem

$$\min_{x' \in \mathcal{X}} \max_{y' \in \mathcal{Y}} \left\{ \phi_\mu(x', y') := \phi(x', y') + \frac{1}{2\mu}\|x' - x\|^2 - \frac{1}{2\mu}\|y' - y\|^2 \right\}. \tag{7}$$

for appropriately chosen $\mu \in (0, \infty)$. (5) can thus be interpreted as solving a particular regularized subproblem. Later we will drop this regularization to arrive at the Lookahead algorithm.

# 6 Last iterate under cohypomonotonicity

As stated in Section 5, we can obtain convergence using the approximate resolvent through Theorem C.1. The convergence is provided in terms of the average, so additional work is needed for a last iterate result. IKM iteration on the approximate resolvent (i.e. $\widetilde{T}_k(z) = J_{\gamma S}(z) + e^k$) becomes,

$$\bar{z}^k = z^k - v^k \quad \text{with} \quad v^k \in \gamma S(\bar{z}^k) \tag{8a}$$
$$z^{k+1} = (1 - \lambda)z^k + \lambda(\bar{z}^k + e^k) \tag{8b}$$

with $\lambda \in (0, 1)$ and $\gamma > 0$ and error $e^k \in \mathbb{R}^d$. Without error, (8) reduces to relaxed proximal point

$$z^{k+1} = (1 - \lambda)z^k + \lambda J_{\gamma S}(z^k) \tag{Relaxed PP}$$

For a last iterate result it remains to argue that the residual $\|J_{\gamma S}(z^k) - z^k\|$ is monotonically decreasing (up to an error we can control). Showing monotonic decrease is fairly straightforward if $\lambda = 1$ (see Lemma E.1 and the associated proof). However, we face additional complication due to the averaging, which is apparent both from the proof and the slightly more complicated error term in the following lemma.

**Lemma 6.1.** *If $S$ is $\rho$-comonotone with $\rho > -\frac{\gamma}{2}$ then* (8) *satisfies for all $z^\star \in \mathrm{zer}\, S$,*

$$\|J_{\gamma S}(z^k) - z^k\|^2 \leq \|J_{\gamma S}(z^{k-1}) - z^{k-1}\|^2 + \delta_k(z^\star)$$

*where $\delta_k(z) := 4\|e^k\|(\|z^{k+1} - z\| + \|z^k - z\|)$.*

The above lemma allows us to obtain last iterate convergence for IKM on the inexact resolvent by combing the lemma with Theorem C.1.

**Algorithm 1** Relaxed approximate proximal point method (RAPP)

---

**Require:** $z^0 \in \mathbb{R}^n$ $\lambda \in (0,1)$, $\gamma \in (\lfloor -2\rho \rfloor_+, 1/L)$
**Repeat** for $k = 0, 1, \dots$ until convergence
  1: $w_k^0 = z^k$
  2: **for all** $t = 0, 1, \dots, \tau - 1$ **do**
  3:      $\xi_{k,t} \sim \mathcal{P}$
  4:      $w_k^{t+1} = (\mathrm{id} + \gamma A)^{-1}(z^k - \gamma \hat{F}_{\sigma_k}(w_k^t, \xi_{k,t}))$
  5: $z^{k+1} = (1 - \lambda)z^k + \lambda w_k^\tau$
**Return** $z^{k+1}$

---

**Theorem 6.2** (Last iterate of inexact resolvent). *Suppose Assumptions 3.2 and 3.4 with $\sigma_k$. Consider the sequence $(z^k)_{k \in \mathbb{N}}$ generated by (8) with $\lambda \in (0,1)$ and $\rho > -\frac{\gamma}{2}$. Then, for all $z^\star \in \mathrm{zer}\, S$,*

$$\mathbb{E}[\|J_{\gamma S}(z^K) - z^K\|^2] \leq \frac{\|z^0 - z^\star\|^2 + \sum_{k=0}^{K-1} \varepsilon_k(z^\star)}{\lambda(1 - \lambda)K} + \frac{1}{K} \sum_{k=0}^{K-1} \sum_{j=k}^{K-1} \delta_j(z^\star),$$

*where $\varepsilon_k(z) := 2\lambda \mathbb{E}[\|e^k\| \|z^k - z\|] + \lambda^2 \mathbb{E}[\|e^k\|^2]$ and $\delta_k(z) := 4\mathbb{E}[\|e^k\|(\|z^{k+1} - z\| + \|z^k - z\|)]$.*

*Remark* 6.3. Notice that the rate in Theorem 6.2 has *no* dependency on $\rho$. Specifically, it gets rid of the factor $\gamma/(\gamma + 2\rho)$ which Gorbunov et al. (2022b, Thm. 3.2) shows is unimprovable for PP. Theorem 6.2 requires that the iterates stays bounded. In Corollary 6.4 we will assume bounded diameter for simplicity, but it is relatively straightforward to show that the iterates can be guaranteed to be bounded by controlling the inexactness (see Lemma E.2).

All that remains to get convergence of the explicit scheme in RAPP, is to expand and simplify the errors $\varepsilon_k(z)$ and $\delta_k(z)$ using the approximation of the resolvent analyzed in Lemma 5.1.

**Corollary 6.4** (Explicit inexact resolvent). *Suppose Assumption 3.2 holds. Consider the sequence $(z^k)_{k \in \mathbb{N}}$ generated by RAPP with deterministic feedback and $\rho > -\frac{\gamma}{2}$. Then, for all $z^\star \in \mathrm{zer}\, S$ with $D := \sup_{j \in \mathbb{N}} \|z^j - z^\star\| < \infty$,*

*(i) with $\tau = \frac{\log K}{\log(1/\gamma L)}$: $\frac{1}{K} \sum_{i=0}^{K-1} \|J_{\gamma S}(z^k) - z^k\|^2 = \mathcal{O}\left( \frac{\|z^0 - z^\star\|^2}{\lambda(1-\lambda)K} + \frac{D^2}{(1-\lambda)K} \right).$*

*(ii) with $\tau = \frac{\log K^2}{\log(1/\gamma L)}$: $\|J_{\gamma S}(z^K) - z^K\|^2 = \mathcal{O}\left( \frac{\|z^0 - z^\star\|^2}{\lambda(1-\lambda)K} + \frac{D^2}{K} + \frac{D^2}{(1-\lambda)K^2}, \right).$*

*Remark* 6.5. Corollary 6.4(ii) implies an oracle complexity of $\mathcal{O}\left( \log(\varepsilon^{-2})\varepsilon^{-1} \right)$ for ensuring that the last iterate satisfies $\|J_{\gamma S}(z^K) - z^K\|^2 \leq \varepsilon$. A stochastic extension is provided in Corollary E.3 by taking the batch size increasing. Notice that RAPP, for $\tau = 2$ inner steps, reduces to EG+ in the unconstrained case where $A \equiv 0$.

# 7 Analysis of Lookahead

The update in RAPP leads to a fairly conservative update in the inner loop, since it corresponds to optimizing a highly regularized subproblem as noted in Section 5. Could we instead replace the optimization procedure with gradient descent ascent (GDA)? If we replace the inner optimization routine we recover what is known as the Lookahead (LA) algorithm

$$
\begin{aligned}
w_k^0 &= z^k \\
w_k^{t+1} &= w_k^t - \gamma F w_k^t \quad \forall t = 0, \dots, \tau - 1 \\
z^{k+1} &= (1 - \lambda)z^k + \lambda w_k^\tau
\end{aligned}
\tag{LA-GDA}
$$

We empirically demonstrate that this scheme can converge for nonmonotone problems for certain choices of parameters (see Figure 2). However, what global guarantees can we provide theoretically?

It turns out that for LA-GDA with two inner steps ($\tau = 2$) we have an affirmative answer. After some algebraic manipulation it is not difficult to see that the update can be simplified as follows

$$z^{k+1} = \tfrac{1}{2}(z^k - 2\lambda\gamma F z^k) + \tfrac{1}{2}(z^k - 2\lambda\gamma F(z^k - \gamma F z^k)). \tag{9}$$

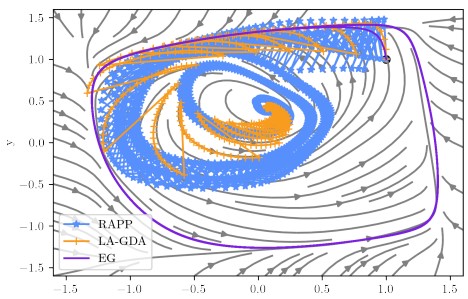 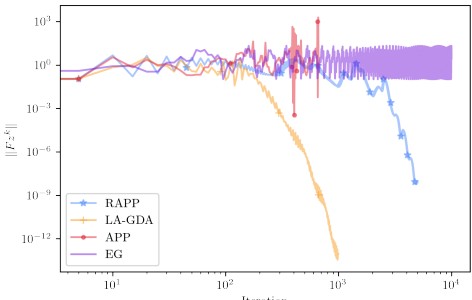

Figure 2: LA-GDA and RAPP can converge for Hsieh et al. (2021, Ex. 5.2). Interestingly, we can set the stepsize $\gamma$ larger than $1/L$ while RAPP remains stable. Approximate proximal point (APP) with the same stepsize diverges (the iterates of APP are deferred to Figure 6). In this example, it is apparent from the rates, that there is a benefit in replacing the conservative inner update in RAPP with GDA in LA-GDA as explored in Section 7.

This is the average of GDA and EG+ (when $\lambda \in (0, 1/2)$). This observation allows us to show convergence under cohypomonotonicity. This positive result for nonmonotone problems partially explains the stabilizing effect of LA-GDA.

**Theorem 7.1.** *Suppose Assumption 3.2 holds. Consider the sequence $(z^k)_{k \in \mathbb{N}}$ generated by LA-GDA with $\tau = 2$, $\gamma \leq 1/L$ and $\lambda \in (0, 1/2)$. Furthermore, suppose that*

$$2\rho > -(1 - 2\lambda)\gamma \quad and \quad 2\rho \geq 2\lambda\gamma - (1 - \gamma^2 L^2)\gamma. \tag{10}$$

*Then, for all $z^\star \in \operatorname{zer} F$,*

$$\frac{1}{K} \sum_{k=0}^{K-1} \|F\bar{z}^k\|^2 \leq \frac{\|z^0 - z^\star\|^2}{\lambda\gamma\big((1 - 2\lambda)\gamma + 2\rho\big)K}. \tag{11}$$

*Remark* 7.2. For $\lambda \to 0$ and $\gamma = c/L$ where $c \in (0, \infty)$, sufficient condition reduces to $\rho \geq -\gamma(1 - \gamma^2 L^2)/2 = -c(1-c^2)/2L$, of which the minimum is attained with $c = 1/\sqrt{3}$, leading to the requirement $\rho \geq -1/3\sqrt{3}L$. A similar statement is possible for $z^k$. Thus, (LA-GDA) improves on the range of $\rho$ compared with EG (see Table 1).

For larger $\tau$, LA-GDA does not necessarily converge (see Figure 3 for a counterexample). We next ask what we would require of the base optimizer to guarantee convergence for any $\tau$. To this end, we replace the inner iteration with some abstract algorithm $\mathrm{Alg} : \mathbb{R}^d \to \mathbb{R}^d$, i.e.

$$
\begin{aligned}
w_k^0 &= z^k \\
w_k^{t+1} &= \mathrm{Alg}(w_k^t) \quad \forall t = 0, ..., \tau - 1 \\
z^{k+1} &= (1 - \lambda)z^k + \lambda w_k^\tau
\end{aligned}
\tag{LA}
$$

Convergence follows from quasi-nonexpansiveness.

**Theorem 7.3.** *Suppose $\mathrm{Alg} : \mathbb{R}^d \to \mathbb{R}^d$ is quasi-nonexpansive. Then $(z^k)_{k \in \mathbb{N}}$ generated by (LA) converges to some $z^\star \in \operatorname{fix} \mathrm{Alg}$.*

*Remark* 7.4. Even though the base optimizer $\mathrm{Alg}$ might not converge (since nonexpansiveness is not sufficient), Theorem 7.3 shows that the outer loop converges. Interestingly, this aligns with the benefit observed in practice of using the outer iteration of Lookahead (see Figure 4).

**Cocoercive** From Theorem 7.3 we almost immediately get converge of LA-GDA for coercive problems since $V = \mathrm{id} - \gamma F$ is nonexpansive iff $\gamma F$ is $1/2$-cocoercive.

**Corollary 7.5.** *Suppose $F$ is $1/L$-cocoercive. Then $(z^k)_{k \in \mathbb{N}}$ generated by LA-GDA with $\gamma \leq 2/L$ converges to some $z^\star \in \operatorname{zer} F$.*

*Remark* 7.6. Corollary 7.5 can trivially be extended to the constrained case by observing that also $V = (\mathrm{id} + \gamma A)^{-1}(\mathrm{id} - \gamma F)$ is nonexpansive when $A$ is maximally monotone. As a special case this captures constrained convex and gradient Lipschitz minimization problems.

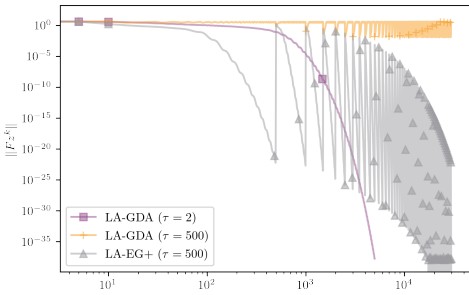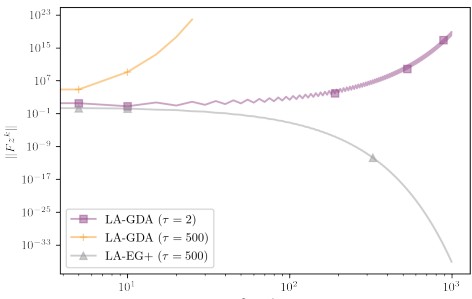

Figure 3: We test the Lookahead variants on Pethick et al. (2022, Ex. 3(iii)) where $\rho \in \left(-1/8L, -1/10L\right)$ (left) and Pethick et al. (2022, Ex. 5) with $\rho = -1/3$ (right). For the left example LA-GDA (provably) converges for $\tau = 2$, but may be nonconvergent for larger $\tau$ as illustrate. Both variants of LA-GDA diverges in the more difficult example on the right, while LA-CEG+ in contrast provably converges. It seems that LA-CEG+ trades off a constant slowdown in the rate for convergence in a larger class.

**Monotone**   When only monotonicity and Lipschitz holds we may instead consider the following extragradient based version of Lookahead (first empirically investigated in Chavdarova et al. (2020))

$$
\begin{aligned}
w_k^0 &= z^k \\
w_k^{t+1} &= \text{EG}(w_t^k) \quad \forall t = 0, ..., \tau - 1 \\
z^{k+1} &= (1 - \lambda)z^k + \lambda w_k^\tau
\end{aligned}
\tag{LA-EG}
$$

where $\text{EG}(z) = z - \gamma F(z - \gamma F z)$. We show in Theorem F.1 that the EG-operator of the inner loop is quasi-nonexpansive, which implies convergence of LA-EG through Theorem 7.3. Theorem F.1 extends even to cases where $A \not\equiv 0$ by utilizing the forward-backward-forward construction of Tseng (1991). This providing the first global convergence guarantee for Lookahead beyond bilinear games.

**Cohypomonotone**   For cohypomonotone problems large $\tau$ may prevent LA-GDA from converging (see Figure 3 for a counterexample). Therefore we propose replacing the inner optimization loop in LA-GDA with the method proposed in (Pethick et al., 2022, Alg. 1). Let $H = \text{id} - \gamma F$. We can write one step of the inner update with $\alpha \in (0, 1)$ as

$$
\text{CEG}^+(w) = w + 2\alpha(H\bar{w} - Hw) \quad \text{with} \quad \bar{w} = (\text{id} + \gamma A)^{-1} Hw.
\tag{12}
$$

The usefulness of the operator $\text{CEG}^+ : \mathbb{R}^d \to \mathbb{R}^d$ comes from the fact that it is quasi-nonexpansive under Assumption 3.2 (see Theorem G.1). Thus, Theorem 7.3 applies even when $F$ is only cohypomonotone if we make the following modification to LA-GDA

$$
\begin{aligned}
w_k^0 &= z^k \\
w_k^{t+1} &= \text{CEG}^+(w_k^t) \quad \forall t = 0, ..., \tau - 1 \\
z^{k+1} &= (1 - \lambda)z^k + \lambda w_k^\tau
\end{aligned}
\tag{LA-CEG+}
$$

In the unconstrained case ($A \equiv 0$) this reduces to using the EG+ algorithm of Diakonikolas et al. (2021) for the inner loop. We have the following convergence guarantee.

**Corollary 7.7.** *Suppose Assumption 3.2 holds. Then $(z^k)_{k \in \mathbb{N}}$ generated by LA-CEG+ with $\lambda \in (0, 1)$, $\gamma \in \left(\lfloor -2\rho \rfloor_+, 1/L\right)$ and $\alpha \in \left(0, 1 + \frac{2\rho}{\gamma}\right)$ converges to some $z^\star \in \text{zer } S$.*

# 8   Experiments

This section demonstrates that linear interpolation can lead to an improvement over common baselines.

**Synthetic examples**   Figures 2 and 3 demonstrate RAPP, LA-GDA and LA-CEG+ on a host of nonmonotone problems (Hsieh et al. (2021, Ex. 5.2), Pethick et al. (2022, Ex. 3(iii)), Pethick et al. (2022, Ex. 5)). See Appendix H.2 for definitions and further details.

Table 2: Adam-based. The combination of Lookahead and extragradient-like methods performs the best.

| | FID | ISC |
|---|---|---|
| Adam | 21.04±2.20 | 7.61±0.15 |
| ExtraAdam | 18.23±1.13 | 7.79±0.08 |
| ExtraAdam+ | 22.94±1.93 | 7.65±0.13 |
| LA-Adam | 17.63±0.65 | 7.86±0.07 |
| LA-ExtraAdam | **15.88±0.67** | 7.97±0.12 |
| LA-ExtraAdam+ | 17.86±1.03 | **8.08±0.15** |

Table 3: GDA-based. Both RAPP and Lookahead increases the scores substantially.

| | FID | ISC |
|---|---|---|
| GDA | 19.36±0.08 | 7.84±0.07 |
| EG | 18.94±0.60 | 7.84±0.02 |
| EG+ | 19.35±4.28 | 7.74±0.44 |
| LA-GDA | **16.87±0.18** | **8.01±0.08** |
| LA-EG | 16.91±0.66 | 7.97±0.12 |
| LA-EG+ | 17.20±0.44 | 7.94±0.11 |
| RAPP | 17.76±0.82 | 7.98±0.08 |

**Image generation** We replicate the experimental setup of Chavdarova et al. (2020); Miyato et al. (2018), which uses hinge version of the non-saturated loss and a ResNet with spectral normalization for the discriminator (see Appendix H.2 for details). To evaluate the performance we rely on the commonly used Inception score (ISC) (Salimans et al., 2016) and Fréchet inception distance (FID) (Heusel et al., 2017) and report the best iterate. We demonstrate the methods on the CIFAR10 dataset (Krizhevsky et al., 2009). The aim is *not* to beat the state-of-the-art, but rather to complement the already exhaustive numerical evidence provided in Chavdarova et al. (2020).

For a fair *computational* comparison we count the number of *gradient computations* instead of iterations $k$ as in Chavdarova et al. (2020). Maybe surprisingly, we find that the extrapolation methods such as EG and RAPP still outperform the baseline, despite having fewer effective iterations. RAPP improves over EG, which suggest that it can be worthwhile to spend more computation on refining the updates at the cost of making fewer updates.

The first experiment we conduct matches the setting of Chavdarova et al. (2020) by relying on the Adam optimizer and using and update ratio of $5:1$ between the discriminator and generator. We find in Table 2 that LA-ExtraAdam+ has the highest ISC (8.08) while LA-ExtraAdam has the lowest FID (15.88). In contrast, we confirm that Adam is unstable while Lookahead prevents divergence as apparent from Figure 4, which is in agreement with Chavdarova et al. (2020). In addition, the *outer* loop of Lookahead achieves better empirical performance, which corroborate the theoretical result (cf. Remark 7.4). Notice that ExtraAdam+ has slow convergence (without Lookahead), which is possibly due to the $1/2$-smaller stepsize.

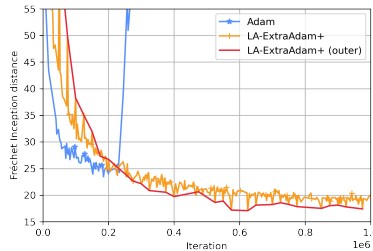

Figure 4: Adam eventually diverges on CIFAR10 while Lookahead is stable with the outer iterate enjoying superior performance.

We additionally simplify the setup by using GDA-based optimizers with an update ratio of $1:1$, which avoids the complexity of diagonal adaptation, gradient history and multiple steps of the discriminator as in the Adam-based experiments. The results are found in Table 3. The learning rates are tuned for GDA and we use those parameters fixed across all other methods. Despite being tuned on GDA, we find that extragradient methods, Lookahead-based methods and RAPP all *still* outperform GDA in terms of FID. The biggest improvement comes from the linear interpolation based methods Lookahead and RAPP (see Figure 8 for further discussion on EG+). Interesting, the Lookahead-based methods are roughly comparable with their Adam variants (Table 2) while GDA even performs better than Adam.

## 9 Conclusion & limitations

We have precisely characterized the stabilizing effect of linear interpolation by analyzing it under cohypomonotonicity. We proved last iterate convergence rates for our proposed method RAPP. The algorithm is double-looped, which introduces a log factor in the rate as mentioned in Remark E.4. It thus remains open whether last iterate is possible using only $\tau = 2$ inner iterations (for which RAPP reduces to EG+ in the unconstrained case). By replacing the inner solver we subsequently rediscovered and analyzed Lookahead using nonexpansive operators. In that regard, we have only dealt with compositions of operators. It would be interesting to further extend the idea to understanding and developing both Federated Averaging and the meta-learning algorithm Reptile (of which Lookahead can be seen as a single client and single task instance respectively), which we leave for future work.

## 10 Acknowledgements

Thanks to Leello Dadi and Stratis Skoulakis for helpful discussion. This work was supported by Google. This work was supported by Hasler Foundation Program: Hasler Responsible AI (project number 21043). This work was supported by the Swiss National Science Foundation (SNSF) under grant number 200021_205011. This project has received funding from the European Research Council (ERC) under the European Union's Horizon 2020 research and innovation programme (grant agreement n° 725594 - time-data).

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
