# Appendix

## Table of Contents

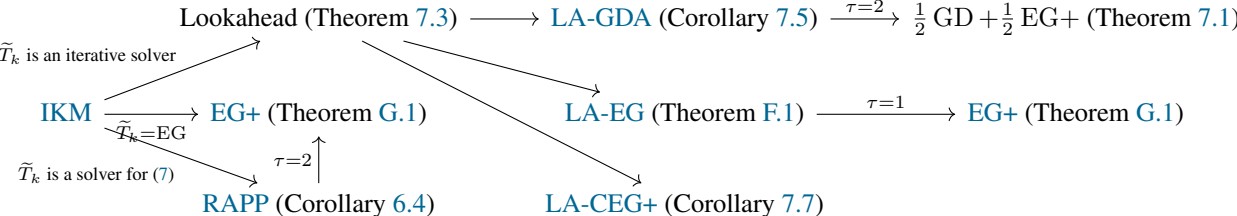

Figure 5: Overview of results and relationship between methods.

# A  Additional related work

**Stochastic feedback**  There are several ways in which a stochastic variant of PP can be devised. *Incremental proximal methods* were pioneered for convex minimization in (Bertsekas, 2011), which uses an implicit update conditioned on the current randomness. Related approaches include Patrascu & Necoara (2017); Bianchi (2015); Patrascu & Irofti (2021); Toulis et al. (2016). Alternatively, (Toulis et al., 2015) assumes noisy access to the *full batch* implicit update in what they call the *proximal Robbins-Monro precedure*, which is similar to the approach taken in Bravo & Cominetti (2022) concerning Krasnoselskii-Mann iterations. Toulis et al. (2015) explicitly approximate the implicit update in the *proximal stochastic fixed-point* algorithm which is closely related to the approximation in Section 5. In the cohypomonotone case it is common to rely on increasing batchsizes (see e.g. (Diakonikolas et al., 2021, Thm. 4.5) and (Lee & Kim, 2021, Thm. 6.1)) similarly to Corollary E.3. Very recently, (Pethick et al., 2023) showed that convergence in stochastic weak MVI (and thus cohypomonotone problems) is possible for an extragradient-type scheme if the Lipschitz conditions are further tightened to a mean-squared smoothness assumption on the stochastic oracles.

**Halpern-type**  Halpern iteration introduced in Halpern (1967), in contrast with IKM, linearly interpolates with the initial point using a time-varying stepsize, i.e. $z^{k+1} = (1 - \lambda_k)z^0 - \lambda_k T z^k$. A $\mathcal{O}(1/k^2)$ convergence rate for the squared fixed point residual was shown in Lieder (2021) for nonexpansive operators. By directly approximating the Halpern iteration, an explicit scheme for monotone problems was later proposed in Diakonikolas (2020), but it suffered a logarithmic factor in the rate. The logarithmic factor was later removed by means of an extragradient variant (Yoon & Ryu, 2021). The scheme was extended to unconstrained cohypomonotone problems in Lee & Kim (2021) and subsequently the constrained case in Cai et al. (2022) while only requiring a single projection.

For a detailed discussion on how Halpern-type methods are not 1-SCLI algorithms see Yoon & Ryu (2021, Appendix E.2), which specifically addresses the anchored extragradient method. The extragradient method, on the other hand, can be written as an 1-SCLI algorithm (c.f. Golowich et al. (2020, Def. 5) and the subsequent discussion). This argument extends to the multistep extragradient construction used in RAPP.

# B  Preliminaries

The distance from $z \in \mathbb{R}^d$ to a set $\mathcal{Z} \subseteq \mathbb{R}^d$ is defined as $\text{dist}(z, \mathcal{Z}) := \min_{z' \in \mathcal{Z}} \|z - z'\|$. The normal cone is defined as $\mathcal{N}_{\mathcal{Z}}(z) := \{ v \mid \langle v, z' - z \rangle \leq 0 \quad \forall z' \in \mathcal{Z} \}$ and the projection as $\mathbf{\Pi}_{\mathcal{Z}}(z) := \min_{w \in \mathcal{Z}} \|z - w\|^2$. We will denote the natural filtration up to iteration $k$ as $\mathcal{F}_k$ and use $\mathbb{E}_k[\cdot] = \mathbb{E}[\cdot \mid \mathcal{F}_k]$.

We restate here some common definitions from monotone and nonexpansive operator for convenience (for further details see Bauschke & Combettes (2017)). An operator $A : \mathbb{R}^d \rightrightarrows \mathbb{R}^n$ maps each point $z \in \mathbb{R}^d$ to a subset $Az \subseteq \mathbb{R}^n$, where the notation $A(z)$ and $Az$ will be used interchangably. We denote the domain of $A$ by $\text{dom}A := \{z \in \mathbb{R}^d \mid Az \neq \emptyset\}$, its graph by $\text{grph}A := \{(z, v) \in \mathbb{R}^d \times \mathbb{R}^n \mid v \in Az\}$. The inverse of $A$ is defined through its graph, $\text{grph}A^{-1} := \{(v, z) \mid (z, v) \in \text{grph}A\}$ and the set of its zeros by $\text{zer}A := \{z \in \mathbb{R}^d \mid 0 \in Az\}$. The set of fixed points is defined as $\text{fix}T := \{z \in \mathbb{R}^d \mid z \in Tz\}$ for the operator $T : \mathbb{R}^d \rightrightarrows \mathbb{R}^d$.

**Definition B.1.** *A single-valued operator $T \colon \mathbb{R}^d \to \mathbb{R}^d$ is said to be*

    *(i) nonexpansive if $\|Tz - Tz'\| \leq \|z - z'\|$    $\forall z, z' \in \mathbb{R}^d$.*

    *(ii) quasi-nonexpansive if $\|Tz - z^\star\| \leq \|z - z^\star\|$    $\forall z \in \mathbb{R}^d$ and $\forall z^\star \in \operatorname{fix} T$.*

    *(iii) firmly nonexpansive if $\|Tz - Tz'\|^2 \leq \|z - z'\|^2 - \|(z - z') - (Tz - Tz')\|^2$    $\forall z, z' \in \mathbb{R}^d$.*

The resolvent operator $J_A := (\operatorname{id} + A)^{-1}$ is firmly nonexpansive (with $\operatorname{dom} J_A = \mathbb{R}^d$) iff $A$ is maximally monotone.

**Definition B.2** ((co)monotonicity Bauschke et al. (2021)). *An operator $A : \mathbb{R}^d \rightrightarrows \mathbb{R}^d$ is called monotone if,*

$$\langle v - v', z - z' \rangle \geq 0 \quad \forall (z, v), (z', v') \in \operatorname{grph} A,$$

*and the operator $A$ is called $\rho$-comonotone (also referred to as $|\rho|$-cohypomonotonicity when $\rho < 0$) if*

$$\langle v - v', z - z' \rangle \geq \rho \|v - v'\|^2 \quad \forall (z, v), (z', v') \in \operatorname{grph} A.$$

*The operator $A$ is* maximally *(co)monotone if no other (co)monotone operator $B$ exists for which* $\operatorname{grph} A \subset \operatorname{grph} B$.

**Definition B.3** (Lipschitz continuity and cocoercivity). *Let $\mathcal{D} \subseteq \mathbb{R}^d$ be a nonempty set. A single-valued operator $A : \mathcal{D} \to \mathbb{R}^n$ is said to be L-Lipschitz continuous if for any $z, z' \in \mathcal{D}$*

$$\|Az - Az'\| \leq L\|z - z'\|,$$

*and $\beta$-cocoercive if*

$$\langle z - z', Az - Az' \rangle \geq \beta \|Az - Az'\|^2.$$

The forward step $H = \operatorname{id} - \gamma F$ is $1/2$-cocoercive when $F$ is Lipschitz continuity and $\gamma$ is sufficiently small.

**Lemma B.4** (Pethick et al. (2022, Lm. A.3(i))). *Suppose Assumption 3.2(ii) holds and $\gamma \leq 1/L$. Then, the mapping $H = \operatorname{id} - \gamma F$ is $1/2$-cocoercive for all $u \in \mathbb{R}^d$. Specifically,*

$$\langle Hz' - Hz, z' - z \rangle \geq \tfrac{1}{2}\|Hz' - Hz\|^2 + \tfrac{1}{2}(1 - \gamma^2 L^2)\|z' - z\|^2 \quad \forall z, z' \in \mathbb{R}^d. \tag{13}$$

*Proof.* By expanding,

$$Hz - Hz' = (z - z') - \gamma(Fz - Fz'). \tag{14}$$

Using (14) we get,

$$\langle Hz' - Hz, z' - z \rangle = \langle Hz' - Hz, Hz' - Hz - \gamma(Fz - Fz') \rangle$$

$$(14) = \tfrac{1}{2}\|Hz' - Hz\|^2 + \tfrac{1}{2}\|z' - z\|^2 - \tfrac{\gamma^2}{2}\|Fz - Fz'\|^2$$

$$(\text{Assumption } 3.2(\text{ii})) \geq \tfrac{1}{2}\|Hz' - Hz\|^2 + \tfrac{1}{2}(1 - \gamma^2 L^2)\|z' - z\|^2 \tag{15}$$

This completes the proof. $\qquad\square$

## B.1    Relationship between weak Minty variational inequilities and cohypomonotonicity

Let $F \colon \mathbb{R}^d \to \mathbb{R}^d$ be a single-valued operator. In the unconstrained case, the weak Minty variational inequality (MVI) with parameter $\rho \in (-1/2L, \infty)$ is defined as

$$\langle Fz, z - z^\star \rangle \geq \rho\|Fz\|^2 \quad \forall z \in \mathbb{R}^d, \forall z^\star \in \operatorname{zer} F. \tag{16}$$

For $\rho < 0$, this condition allows the operator $-Fz$ to point away from the solution set as illustrated in Figure 1.

Notice that since $z^\star \in \operatorname{zer} F$ we could equivalently write

$$\langle Fz - Fz^\star, z - z^\star \rangle \geq \rho\|Fz - Fz^\star\|^2. \quad \forall z \in \mathbb{R}^d \tag{17}$$

In contrast, $\rho$-comonotonicity of $F$ states that the above condition should hold for all pairs of point in the domain, i.e.

$$\langle Fz - Fz', z - z' \rangle \geq \rho\|Fz - Fz'\|^2 \quad \forall z, z' \in \mathbb{R}^d.$$

For $\rho < 0$, $\rho$-comonotonicity is also referred to as $|\rho|$-cohypomonotonicity. We say that the weak MVI is a *star-variant* of comonotonicity. This is analogue to the relationship between convexity and star-convexity.

For simplicity we state all results in terms of comonotonicity. However, note that *almost all results in this paper trivially extends to the more relaxed notion of weak MVI*. The only exception is the last iterate rates in Theorem 6.2 which relies on cohypomonotonicity to prove monotonic decrease through Lemma 6.1.

# C  Proofs for Section 4 (Inexact Krasnosel'skiĭ-Mann iterations)

The IKM iteration is well studied (see Combettes (2001)). The following result refurbishes sub-results of Combettes (2001, Prop. 4.2) to establish a rate of convergence under potentially stochastic feedback.

**Theorem C.1** (Convergence of IKM). *Suppose $T : \mathbb{R}^d \to \mathbb{R}^d$ is quasi-nonexpansive. Consider the sequence $(z^k)_{k \in \mathbb{N}}$ generated by IKM with $\lambda \in (0,1)$. Then, for all $z^\star \in \text{fix } T$*

$$\frac{1}{K} \sum_{k=0}^{K-1} \mathbb{E} \|Tz^k - z^k\|^2 \leq \frac{\|z^0 - z^\star\|^2 + \sum_{k=0}^{K-1} \varepsilon_k(z^\star)}{\lambda(1-\lambda)K}. \tag{18}$$

*where $\varepsilon_k(z) = 2\lambda\mathbb{E}[\|e^k\|\|z^k - z\|] + \lambda^2\mathbb{E}[\|e^k\|^2]$. Furthermore, $z^k \to z^\star$ a.s. as long as $\sum_{k=0}^{\infty} \varepsilon_k(z^\star) < \infty$.*

*Remark* C.2. Notice that the optimal choice of $\lambda$ in the upper bound is $\lambda = 0.5$, which is the default used for the Lookahead algorithm in both for minimax problems (Chavdarova et al., 2020) and minimization (Zhang et al., 2019) (see Section 7 for a treatment of Lookahead).

*Proof.* We will denote the natural filtration up to iteration $k$ as $\mathcal{F}_k$ and use $\mathbb{E}_k[\cdot] = \mathbb{E}[\cdot \mid \mathcal{F}_k]$. Consider one exact step

$$s^k = (1-\lambda)z^k + \lambda Tz^k \tag{19}$$

Then

$$\|s^k - z^\star\|^2 = (1-\lambda)\|z^k - z^\star\|^2 + \lambda\|Tz^k - z^\star\|^2 - \lambda(1-\lambda)\|Tz^k - z^k\|^2$$

$$\text{(quasi-nonexpansive)} \leq (1-\lambda)\|z^k - z^\star\|^2 + \lambda\|z^k - z^\star\|^2 - \lambda(1-\lambda)\|Tz^k - z^k\|^2$$

$$= \|z^k - z^\star\|^2 - \lambda(1-\lambda)\|Tz^k - z^k\|^2 \tag{20}$$

So

$$\|s^k - z^\star\| \leq \|z^k - z^\star\| \tag{21}$$

By using triangle inequality and the update rule IKM we have,

$$\mathbb{E}_k[\|z^{k+1} - z^\star\|^2] \leq \mathbb{E}_k[(\|s^k - z^\star\| + \lambda\|e^k\|)^2]$$

$$= \|s^k - z^\star\|^2 + 2\lambda\mathbb{E}_k[\|e^k\|]\|s^k - z^\star\| + \lambda^2\mathbb{E}_k[\|e^k\|^2]$$

$$(21) \leq \|s^k - z^\star\|^2 + 2\lambda\mathbb{E}_k[\|e^k\|]\|z^k - z^\star\| + \lambda^2\mathbb{E}_k[\|e^k\|^2]$$

$$(20) \leq \|z^k - z^\star\|^2 - \lambda(1-\lambda)\|Tz^k - z^k\|^2 + 2\lambda\mathbb{E}_k[\|e^k\|]\|z^k - z^\star\| + \lambda^2\mathbb{E}_k[\|e^k\|^2]. \tag{22}$$

Using law of total expectation and telescoping obtains the claimed rate. The claimed asymptotic result follows from the Robbins-Siegmund supermartingale theorem (Bertsekas, 2011, Prop. 2). This completes the proof. □

# D  Proofs for Section 5 (Approximating the resolvent)

**Lemma 5.1.** *Suppose Assumptions 3.2(i), 3.2(ii) and 3.4. Given $z \in \mathbb{R}^d$, the iterates generated by (5) with $\gamma \in (0, 1/L)$ converges to a neighborhood linearly, i.e.,*

$$\mathbb{E}[\|w^\tau - J_{\gamma S}(z)\|^2] \leq (\gamma L)^{2\tau}\|w^0 - w^\star\|^2 + \frac{\gamma^2}{(1-\gamma L)^2}\sigma^2. \tag{6}$$

*Proof.* Let $\zeta^t = Fw^t - \hat{F}_\sigma(w^t, \xi_t)$. Then the stochastic update in (5) can be written as

$$w^{t+1} = (\mathrm{id} + \gamma A)^{-1}(z - \gamma Fw^t + \gamma\zeta^t) \tag{23}$$

Let $w^\star \in \mathrm{fix}\, Q_z$ such that

$$\|w^{t+1} - w^\star\|^2 = \|w^{t+1} - Q_z(w^\star)\|^2. \tag{24}$$

Due to (firmly) nonexpansiveness of $(\mathrm{id} + \gamma A)^{-1}$ when $A$ is maximally monotone we can go on as

$$
\begin{aligned}
\|w^{t+1} - Q_z(w^\star)\|^2 &= \|(\mathrm{id} + \gamma A)^{-1}(z - \gamma Fw^t + \gamma\zeta^t) - (\mathrm{id} + \gamma A)^{-1}(z - \gamma Fw^\star)\|^2 \\
&\leq \|(z - \gamma Fw^t + \gamma\zeta^t) - (z - \gamma Fw^\star)\|^2 \\
&= \gamma^2\|Fw^t - Fw^\star\|^2 + \gamma^2\|\zeta^t\|^2 + 2\gamma^2\,\langle\zeta^t, Fw^\star - Fw^t\rangle \\
&\leq \gamma^2 L^2\|w^t - w^\star\|^2 + \gamma^2\|\zeta^t\|^2 + 2\gamma^2\,\langle\zeta^t, Fw^\star - Fw^t\rangle
\end{aligned}
\tag{25}
$$

where the last inequality follows from Lipschitz continuity of $F$.

Taking expectation and using unbiasedness and bounded variance from Assumption 3.4 we get

$$\mathbb{E}\big[\|w^{t+1} - w^\star\|^2 \mid \mathcal{F}_t\big] \leq \gamma^2 L^2\|w^t - w^\star\|^2 + \gamma^2\sigma^2 \tag{26}$$

By law of total expectation

$$
\begin{aligned}
\mathbb{E}\big[\|w^\tau - w^\star\|\big] &\leq \gamma^2 L^2 \mathbb{E}\big[\|w^{\tau-1} - w^\star\|^2\big] + \gamma^2\sigma^2 \\
&\leq \gamma^4 L^4 \mathbb{E}\big[\|w^{\tau-2} - w^\star\|^2\big] + \gamma^2(1 + \gamma^2 L^2)\sigma^2 \\
&\leq \cdots \leq (\gamma L)^{2\tau}\mathbb{E}\big[\|w^0 - w^\star\|^2\big] + \gamma^2\sigma^2\sum_{t=0}^{\tau-1}(\gamma L)^{2t} \\
&\leq (\gamma L)^{2\tau}\|w^0 - w^\star\|^2 + \tfrac{\gamma^2}{(1-\gamma L)^2}\sigma^2
\end{aligned}
$$

where the last inequality follows from $\sum_{t=0}^\infty a^t = \frac{1}{1-a}$ when $a < 1$.

By construction $\mathrm{fix}\, Q_z = \{J_{\gamma S}(z)\}$ which completes the proof. $\qquad\square$

# E  Proofs for Section 6 (Last iterate under cohypomonotonicity)

**Lemma 6.1.** *If $S$ is $\rho$-comonotone with $\rho > -\frac{\gamma}{2}$ then (8) satisfies for all $z^\star \in \mathrm{zer}\, S$,*

$$\|J_{\gamma S}(z^k) - z^k\|^2 \leq \|J_{\gamma S}(z^{k-1}) - z^{k-1}\|^2 + \delta_k(z^\star)$$

*where* $\delta_k(z) := 4\|e^k\|(\|z^{k+1} - z\| + \|z^k - z\|)$.

*Proof.* Rearranging the update (8b) and subsequently using (8a),

$$z^k - z^{k+1} = \lambda(z^k - \bar{z}^k - e^k) = \lambda(v^k - e^k). \tag{27}$$

Since $\gamma S$ is $\frac{1}{2}$-cohypomonotone

$$
\begin{aligned}
-\tfrac{1}{2}\|v^k - v^{k+1}\|^2 &\leq \langle v^k - v^{k+1}, \bar{z}^k - \bar{z}^{k+1}\rangle \\
\text{(8a)} &= \langle v^k - v^{k+1}, z^k - v^k - (z^{k+1} - v^{k+1})\rangle \\
&= \langle v^k - v^{k+1}, z^k - z^{k+1}\rangle - \|v^k - v^{k+1}\|^2 \\
\text{(27)} &= \lambda\,\langle v^k - v^{k+1}, v^k - e^k\rangle - \|v^k - v^{k+1}\|^2 \\
&= \lambda\|v^k\|^2 - \lambda\,\langle v^{k+1}, v^k\rangle - \|v^k - v^{k+1}\|^2 + \lambda\,\langle v^{k+1} - v^k, e^k\rangle
\end{aligned}
\tag{28}
$$

Rearranging

$$
\begin{aligned}
0 &\leq \lambda\|v^k\|^2 - \lambda\,\langle v^{k+1}, v^k\rangle - \tfrac{1}{2}\|v^k - v^{k+1}\|^2 + \lambda\,\langle v^{k+1} - v^k, e^k\rangle \\
&\leq \lambda\|v^k\|^2 - \lambda\,\langle v^{k+1}, v^k\rangle - \tfrac{\lambda}{2}\|v^k - v^{k+1}\|^2 + \lambda\,\langle v^{k+1} - v^k, e^k\rangle \\
&= \lambda\|v^k\|^2 - \tfrac{\lambda}{2}\|v^k\|^2 - \tfrac{\lambda}{2}\|v^{k+1}\|^2 + \lambda\,\langle v^{k+1} - v^k, e^k\rangle
\end{aligned}
\tag{29}
$$

where the second inequality follows from observing that $\frac{1}{2} > \frac{\lambda}{2}$ since $\lambda \in (0,1)$. It remain to bound the error term. Since $\gamma S$ is $1/2$-cohypomonotone the resolvent $J_{\gamma S}$ is nonexpansive. Thus,

$$\|\bar{z}^k - z^\star\| \le \|z^k - z^\star\|. \tag{30}$$

Using Cauchy-Schwarz and the triangle inequality,

$$
\begin{aligned}
\langle v^{k+1} - v^k, e^k \rangle &\le \|e^k\| \|v^{k+1} - v^k\| \\
&\le \|e^k\|(\|v^{k+1}\| + \|v^k\|) \\
&= \|e^k\|(\|\bar{z}^{k+1} - z^{k+1}\| + \|\bar{z}^k - z^k\|) \\
&\le \|e^k\|(\|z^{k+1} - z^\star\| + \|z^k - z^\star\| + \|\bar{z}^{k+1} - z^\star\| + \|\bar{z}^k - z^\star\|\|) \\
(30) \quad &\le 2\|e^k\|(\|z^{k+1} - z^\star\| + \|z^k - z^\star\|)
\end{aligned}
\tag{31}
$$

Combining (29) and (31),

$$\tfrac{1}{2}\|v^{k+1}\|^2 \le \tfrac{1}{2}\|v^k\|^2 + 2\|e^k\|(\|z^{k+1} - z^\star\| + \|z^k - z^\star\|). \tag{32}$$

Substituting in the resolvent using (8a) completes the proof. $\qquad\square$

The proof of Lemma 6.1 simplifies for $\lambda = 1$. Consider one application of the inexact resolvent with error $e \in \mathbb{R}^d$,

$$z' = J_{\gamma S}(z) + e, \tag{33}$$

where $\lambda \in (0,1)$ and $\gamma > 0$.

**Lemma E.1.** *If $S$ is $\rho$-comonotone with $\rho > -\frac{\gamma}{2}$ then (33) satisfies $\|J_{\gamma S}(z') - z'\| \le \|J_{\gamma S}(z) - z\| + 2\|e\|$.*

*Proof.* Since $\gamma S$ is $1/2$-cohypomonotone the resolvent $J_{\gamma S}$ is nonexpansive. Thus,

$$
\begin{aligned}
\|J_{\gamma S}(z') - z'\| &= \|J_{\gamma S}(z') - J_{\gamma S}(z) - e\| \\
(\text{triangle ineq.}) &\le \|J_{\gamma S}(z') - J_{\gamma S}(z)\| + \|e\| \\
(\text{nonexpansive}) &\le \|z' - z\| + \|e\| \\
(\text{triangle ineq.}) &\le \|J_{\gamma S}(z) - z\| + 2\|e\|
\end{aligned}
$$

This completes the proof. $\qquad\square$

Furthermore, the iterates of (8) are bounded in the following sense.

**Lemma E.2.** *Consider the sequence $(z^k)_{k \in \mathbb{N}}$ generated by (8) with $\lambda \in (0,1)$ and $\rho > -\frac{\gamma}{2}$. Then for any $z^\star \in \operatorname{zer} S$,*

$$\|z^{k+1} - z^\star\| \le \|z^0 - z^\star\| + \lambda \sum_{j=0}^{k} \|e^j\|. \tag{34}$$

*Proof.* Since $\gamma S$ is $1/2$-cohypomonotone the resolvent $J_{\gamma S}$ is nonexpansive. Thus,

$$\|\bar{z}^k - z^\star\| \le \|z^k - z^\star\|. \tag{35}$$

We use the update rule

$$
\begin{aligned}
\|z^{k+1} - z^\star\| &= \|(1-\lambda)z^k + \lambda(\bar{z}^k + e^k) - z^\star\| \\
&\le \|(1-\lambda)(z^k - z^\star) + \lambda(\bar{z}^k - z^\star)\| + \lambda\|e^k\| \\
&\le (1-\lambda)\|z^k - z^\star\| + \lambda\|\bar{z}^k - z^\star\| + \lambda\|e^k\| \\
(35) \quad &\le \|z^k - z^\star\| + \lambda\|e^k\|
\end{aligned}
\tag{36}
$$

By recursively applying (36) we obtain the claim. $\qquad\square$

**Theorem 6.2** (Last iterate of inexact resolvent). *Suppose Assumptions 3.2 and 3.4 with $\sigma_k$. Consider the sequence $(z^k)_{k\in\mathbb{N}}$ generated by (8) with $\lambda \in (0,1)$ and $\rho > -\frac{\gamma}{2}$. Then, for all $z^\star \in \operatorname{zer} S$,*

$$\mathbb{E}[\|J_{\gamma S}(z^K) - z^K\|^2] \leq \frac{\|z^0 - z^\star\|^2 + \sum_{k=0}^{K-1} \varepsilon_k(z^\star)}{\lambda(1-\lambda)K} + \frac{1}{K} \sum_{k=0}^{K-1} \sum_{j=k}^{K-1} \delta_j(z^\star),$$

*where $\varepsilon_k(z) := 2\lambda\mathbb{E}[\|e^k\|\|z^k - z\|] + \lambda^2\mathbb{E}[\|e^k\|^2]$ and $\delta_k(z) := 4\mathbb{E}[\|e^k\|(\|z^{k+1} - z\| + \|z^k - z\|)]$.*

*Proof.* By taking $T = J_{\gamma S}$ in Theorem C.1 we have

$$\frac{1}{K} \sum_{k=0}^{K-1} \mathbb{E}[\|J_{\gamma S}(z^k) - z^k\|^2] \leq \frac{\|z^0 - z^\star\|^2 + \sum_{k=0}^{K-1} \varepsilon_k(z^\star)}{\lambda(1-\lambda)K}. \tag{37}$$

From Lemma 6.1 (and law of total expectation) we obtain,

$$K\mathbb{E}[\|J_{\gamma S}(z^K) - z^K\|^2] \leq \sum_{k=0}^{K-1} \mathbb{E}[\|J_{\gamma S}(z^k) - z^k\|^2] + \sum_{k=0}^{K-1} \sum_{j=k}^{K-1} \delta_j(z^\star). \tag{38}$$

Dividing by $K$ and combining with (37) yields the rate. Noticing that fix $J_{\gamma S} = \operatorname{zer} S$ completes the proof. $\square$

**Corollary E.3** (Explicit inexact stochastic resolvent). *Suppose Assumptions 3.2 and 3.4 with $\sigma_k$ for all $k \in \mathbb{N}$. Consider the sequence $(z^k)_{k\in\mathbb{N}}$ generated by RAPP with $\rho > -\frac{\gamma}{2}$. Then, for all $z^\star \in \operatorname{zer} S$ with $D := \sup_{j\in\mathbb{N}} \|z^j - z^\star\| < \infty$,*

*(i) with $\sigma_k^2 = \sigma_0^2/k^2$ and $\tau = \frac{\log K}{\log(1/\gamma L)}$,*

$$\frac{1}{K} \sum_{i=0}^{K-1} \mathbb{E}[\|J_{\gamma S}(z^k) - z^k\|^2] \leq \frac{\|z^0 - z^\star\|^2}{\lambda(1-\lambda)K} + \mathcal{O}\Big(\max\Big\{\frac{D^2}{(1-\lambda)K}, \frac{\gamma D\sigma_0}{(1-\gamma L)(1-\lambda)K}\Big\}$$

$$+ \frac{\lambda D^2}{(1-\lambda)K} + \frac{\lambda\gamma^2\sigma_0^2}{(1-\gamma L)^2(1-\lambda)K^2}\Big).$$

*(ii) with $\sigma_k^2 = \sigma_0^2/k^3$ and $\tau = \frac{\log K^2}{\log(1/\gamma L)}$,*

$$\mathbb{E}[\|J_{\gamma S}(z^K) - z^K\|^2] \leq \frac{\|z^0 - z^\star\|^2}{\lambda(1-\lambda)K} + \mathcal{O}\Big(\max\{\frac{D^2}{K}, \frac{8\gamma D\sigma_0}{(1-\gamma L)\sqrt{K}}\}\Big)$$

$$+ \mathcal{O}\Big(\max\Big\{\frac{D^2}{(1-\lambda)K^2}, \frac{2\gamma D\sigma_0}{(1-\gamma L)(1-\lambda)K^{3/2}}\Big\} \tag{39}$$

$$+ \frac{\lambda D^2}{(1-\lambda)K^2} + \frac{\lambda\gamma^2\sigma_0^2}{(1-\gamma L)^2(1-\lambda)K^3}\Big)$$

*Remark* E.4. The assumption on the noise $\sigma_k^2 = \sigma_0^2/n_k$ can be achieved by taking the batch size as $n_k$, i.e.

$$\hat{F}_{\sigma_k}(z, \xi) = \frac{1}{n_k} \sum_{i=0}^{n_k} \hat{F}_{\sigma_0}(z, \xi_i). \tag{40}$$

This is clear by simple computation. Observe that the random variable $X_i := \hat{F}_\sigma(z, \xi_i) - Fz$ is i.i.d. with $\operatorname{Var}(X_i) = \sigma^2$. Then, the average, $\overline{X}_n = \frac{1}{n}(X_1 + \cdots + X_n)$, has a variance as follows

$$\operatorname{Var}(\overline{X}_n) = \operatorname{Var}(\tfrac{1}{n}(X_1 + \cdots + X_n)) = \frac{1}{n^2}\operatorname{Var}(X_1 + \cdots + X_n) = \frac{n\sigma^2}{n^2} = \frac{\sigma^2}{n}.$$

We note that increasing batch size might be unfavorable in some applications, but the alternative of diminishing stepsize only leads to only asymptotic convergence of the last iterate (as in e.g. Pethick et al. (2023)).

*Proof.* The theorem follows from combing Lemma 5.1 with Theorems 6.2 and C.1. Invoke Theorems 6.2 and C.1 with $e^k = w_k^\tau - J_{\gamma S}(z^k)$ and $\sigma = \sigma_k$ and note that the error $e^k$ can be bounded through Lemma 5.1 as

$$\mathbb{E}_k[\|e^k\|^2] = \|w_k^\tau - J_{\gamma S}(z^k)\|^2 \le \gamma^{2\tau} L^{2\tau} \|w_k^0 - J_{\gamma S}(z^k)\|^2 + \frac{\gamma^2}{(1-\gamma L)^2}\sigma_k^2$$

$$= \gamma^{2\tau} L^{2\tau} \|z^k - J_{\gamma S}(z^k)\|^2 + \frac{\gamma^2}{(1-\gamma L)^2}\sigma_k^2. \tag{41}$$

The former term can in turn be bounded through the triangle inequality

$$\|z^k - J_{\gamma S}(z^k)\| \le \|z^k - z^\star\| + \|J_{\gamma S}(z^k) - z^\star\| \le 2\|z^k - z^\star\| \le 2D. \tag{42}$$

with $D := \sup_{j \in \mathbb{N}} \|z^j - z^\star\|$ and where the second last inequality follows from $z^\star \in \text{fix } J_{\gamma S}$ and nonexpansiveness of $J_{\gamma S}$. Plugging into (41) we have,

$$\mathbb{E}_k[\|e^k\|^2] \le 4\gamma^{2\tau} L^{2\tau} D^2 + \frac{\gamma^2}{(1-\gamma L)^2}\sigma_k^2, \tag{43}$$

and

$$\mathbb{E}_k[\|e^k\|] \le \sqrt{4\gamma^{2\tau} L^{2\tau} D^2 + \frac{\gamma^2}{(1-\gamma L)^2}\sigma_k^2} \le \max\{2\gamma^\tau L^\tau D, \tfrac{\gamma}{1-\gamma L}\sigma_k\}. \tag{44}$$

Substituting into the expression of $\delta_k(z^\star)$ and $\varepsilon_k(z^\star)$ yields,

$$\delta_k(z^\star) \le \max\{16\gamma^\tau L^\tau D^2, \tfrac{8\gamma}{1-\gamma L}\sigma_k D\}$$

$$\varepsilon_k(z^\star) \le \max\{4\lambda\gamma^\tau L^\tau D^2, \tfrac{2\lambda\gamma}{1-\gamma L}\sigma_k D\} + 4\lambda^2\gamma^{2\tau} L^{2\tau} D^2 + \frac{\lambda^2\gamma^2}{(1-\gamma L)^2}\sigma_k^2.$$

Consequently, with the choice $\sigma_k^2 = \sigma_0^2/k^2$,

$$\frac{\sum_{k=0}^{K-1} \varepsilon_k(z^\star)}{\lambda(1-\lambda)K} \le \max\left\{\frac{4\gamma^\tau L^\tau D^2}{1-\lambda}, \frac{2\gamma D\sigma_0}{(1-\gamma L)(1-\lambda)K}\right\} + \frac{4\lambda\gamma^{2\tau} L^{2\tau} D^2}{1-\lambda} + \frac{\lambda\gamma^2\sigma_0^2}{(1-\gamma L)^2(1-\lambda)K^2}. \tag{45}$$

We ideally want the terms involving $\tau$ to be of order $\mathcal{O}(1/K)$.

$$1/a^\tau = 1/K \iff a^\tau = K \iff \tau \log a = \log K \iff \tau = \frac{\log K}{\log a} \tag{46}$$

Choosing $a = 1/\gamma L$ it thus suffice to pick $\tau = \frac{\log K}{\log(1/\gamma L)}$ in order to have $\gamma^\tau L^\tau = 1/K$. Plugging into the average iterate result of Theorem C.1 yields the claim in Corollary E.3(i).

Additionally, with the choice $\sigma_k^2 = \sigma_0^2/k^3$,

$$\frac{\sum_{k=0}^{K-1} \varepsilon_k(z^\star)}{\lambda(1-\lambda)K} \le \max\left\{\frac{4\gamma^\tau L^\tau D^2}{1-\lambda}, \frac{2\gamma D\sigma_0}{(1-\gamma L)(1-\lambda)K^{3/2}}\right\} + \frac{4\lambda\gamma^{2\tau} L^{2\tau} D^2}{1-\lambda} + \frac{\lambda\gamma^2\sigma_0^2}{(1-\gamma L)^2(1-\lambda)K^3}$$

$$\frac{1}{K}\sum_{k=0}^{K-1}\sum_{j=k}^{K-1}\delta_j(z^\star) \le \max\{K16\gamma^\tau L^\tau D^2, \tfrac{8\gamma D\sigma_0}{(1-\gamma L)\sqrt{K}}\}. \tag{47}$$

In order for the terms involving $\tau$ to be of order $\mathcal{O}(1/K)$ we need $\tau$ to be slightly larger.

$$K/a^\tau = 1/K \iff a^\tau = K^2 \iff \tau \log a = \log K^2 \iff \tau = \frac{\log K^2}{\log a} \tag{48}$$

Choosing $a = 1/\gamma L$ it thus suffice to pick $\tau = \frac{\log K^2}{\log(1/\gamma L)}$ in order to have $K\gamma^\tau L^\tau = 1/K$. Plugging (47) into the last iterate result of Theorem 6.2 completes the proof. $\square$

**Corollary 6.4** (Explicit inexact resolvent). *Suppose Assumption 3.2 holds. Consider the sequence $(z^k)_{k\in\mathbb{N}}$ generated by RAPP with deterministic feedback and $\rho > -\frac{\gamma}{2}$. Then, for all $z^\star \in \text{zer } S$ with $D := \sup_{j\in\mathbb{N}} \|z^j - z^\star\| < \infty$,*

(i) *with $\tau = \frac{\log K}{\log(1/\gamma L)}$: $\frac{1}{K}\sum_{i=0}^{K-1}\|J_{\gamma S}(z^k) - z^k\|^2 = \mathcal{O}\left(\frac{\|z^0 - z^\star\|^2}{\lambda(1-\lambda)K} + \frac{D^2}{(1-\lambda)K}\right)$.*

(ii) *with $\tau = \frac{\log K^2}{\log(1/\gamma L)}$: $\|J_{\gamma S}(z^K) - z^K\|^2 = \mathcal{O}\left(\frac{\|z^0 - z^\star\|^2}{\lambda(1-\lambda)K} + \frac{D^2}{K} + \frac{D^2}{(1-\lambda)K^2},\right)$.*

*Proof.* The claim follows directly from Corollary E.3 as a special case with $\sigma_0 = 0$. $\square$

# F   Proofs for Section 7 (Analysis of Lookahead)

**Theorem 7.1.** *Suppose Assumption 3.2 holds. Consider the sequence $(z^k)_{k\in\mathbb{N}}$ generated by LA-GDA with $\tau = 2$, $\gamma \le 1/L$ and $\lambda \in (0, 1/2)$. Furthermore, suppose that*

$$2\rho > -(1-2\lambda)\gamma \quad \text{and} \quad 2\rho \ge 2\lambda\gamma - (1-\gamma^2 L^2)\gamma. \tag{10}$$

*Then, for all $z^\star \in \mathrm{zer}\, F$,*

$$\frac{1}{K}\sum_{k=0}^{K-1}\|F\bar{z}^k\|^2 \le \frac{\|z^0 - z^\star\|^2}{\lambda\gamma\big((1-2\lambda)\gamma + 2\rho\big)K}. \tag{11}$$

*Proof.* For $\tau = 2$ we can write (LA-GDA) as

$$\begin{aligned}
z^{k+1/3} &= z^k - \gamma F z^k \\
z^{k+2/3} &= z^{k+1/3} - \gamma F z^{k+1/3} \\
z^{k+1} &= (1-\lambda)z^k + \lambda z^{k+2/3}
\end{aligned} \tag{49}$$

The proof relies on the simplified form of the update rule (9), which can be obtain as follows

$$\begin{aligned}
z^{k+1} &= (1-\lambda)z^k + \lambda z^{k+2/3} \\
&= (1-\lambda)z^k + \lambda(z^{k+1/3} - \gamma F z^{k+1/3}) \\
&= (1-\lambda)z^k + \lambda(z^k - \gamma F z^k - \gamma F z^{k+1/3}) \\
&= z^k - \lambda\gamma F z^k - \lambda\gamma F(z^k - \gamma F z^k) \\
&= \tfrac{1}{2}(z^k - 2\lambda\gamma F z^k) + \tfrac{1}{2}(z^k - 2\lambda\gamma F(z^k - \gamma F z^k)).
\end{aligned} \tag{50}$$

Define the following operators with $\beta = 2\lambda$

$$\begin{aligned}
\mathrm{EG}^+(z) &= z - \beta\gamma F(z - \gamma F z) & \text{(51a)} \\
\mathrm{GDA}(z) &= z - \beta\gamma F z & \text{(51b)}
\end{aligned}$$

Then, using (50), LA-GDA with $\tau = 2$ can be written as

$$z^{k+1} = \tfrac{1}{2}\,\mathrm{GDA}(z^k) + \tfrac{1}{2}\,\mathrm{EG}^+(z^k) \tag{52}$$

One step of the update can be bounded as

$$\|z^{k+1} - z^\star\|^2 = \|\tfrac{1}{2}\,\mathrm{GDA}(z^k) + \tfrac{1}{2}\,\mathrm{EG}^+(z^k) - z^\star\|^2 \le \tfrac{1}{2}\|\mathrm{GDA}(z^k) - z^\star\|^2 + \tfrac{1}{2}\|\mathrm{EG}^+(z^k) - z^\star\|^2, \tag{53}$$

where we have used Young's inequality. The first term can be expanded

$$\|\mathrm{GDA}(z^k) - z^\star\|^2 = \|z^k - z^\star\|^2 + \beta^2\gamma^2\|F z^k\|^2 - 2\beta\gamma\langle F z^k, z^k - z^\star\rangle \tag{54}$$

For the second term of (53) we will need to bound the following inner product

$$\begin{aligned}
\langle \gamma F\bar{z}^k, z^k - \bar{z}^k\rangle &= \tfrac{\gamma^2}{2}\|F\bar{z}^k\|^2 - \tfrac{1}{2}\|\gamma F\bar{z}^k - (z^k - \bar{z}^k)\|^2 + \tfrac{1}{2}\|\bar{z}^k - z^k\|^2 \\
\text{(51a)} &= \tfrac{\gamma^2}{2}\|F\bar{z}^k\|^2 - \tfrac{\gamma^2}{2}\|F\bar{z}^k - F z^k\|^2 + \tfrac{1}{2}\|\bar{z}^k - z^k\|^2 \\
\text{(Assumption 3.2(ii))} &\ge \tfrac{\gamma^2}{2}\|F\bar{z}^k\|^2 + \tfrac{1}{2}(1 - \gamma^2 L^2)\|\bar{z}^k - z^k\|^2.
\end{aligned} \tag{55}$$

Consequently,

$$\begin{aligned}
\gamma\langle F\bar{z}^k, z^k - z^\star\rangle &= \gamma\langle F\bar{z}^k, \bar{z}^k - z^\star\rangle + \gamma\langle F\bar{z}^k, z^k - \bar{z}^k\rangle \\
\text{(55)} &\le \gamma\langle F\bar{z}^k, \bar{z}^k - z^\star\rangle - \tfrac{\gamma^2}{2}\|F\bar{z}^k\|^2 - \tfrac{1}{2}(1 - \gamma^2 L^2)\|\bar{z}^k - z^k\|^2.
\end{aligned} \tag{56}$$

Finally,

$$\begin{aligned}
\|\mathrm{EG}^+(z^k) - z^\star\|^2 &= \|z^k - z^\star\|^2 + \beta^2\gamma^2\|F\bar{z}^k\|^2 - 2\beta\gamma\langle F\bar{z}^k, z^k - z^\star\rangle \\
\text{(56)} &\le \|z^k - z^\star\|^2 - \beta(1-\beta)\gamma^2\|F\bar{z}^k\|^2 - \beta(1-\gamma^2 L^2)\|\bar{z}^k - z^k\|^2 - 2\beta\gamma\langle F\bar{z}^k, \bar{z}^k - z^\star\rangle \\
\text{(51a)} &= \|z^k - z^\star\|^2 - \beta(1-\beta)\gamma^2\|F\bar{z}^k\|^2 - \beta(1-\gamma^2 L^2)\gamma^2\|F z^k\|^2 - 2\beta\gamma\langle F\bar{z}^k, \bar{z}^k - z^\star\rangle
\end{aligned} \tag{57}$$

Using (54) and (57) in (53), we have

$$2\|z^{k+1} - z^\star\|^2 \le 2\|z^k - z^\star\|^2 + \beta^2\gamma^2\|Fz^k\|^2 - 2\beta\gamma\langle Fz^k, z^k - z^\star\rangle$$
$$- \beta(1-\beta)\gamma^2\|F\bar{z}^k\|^2 - \beta(1-\gamma^2L^2)\gamma^2\|Fz^k\|^2 - 2\beta\gamma\langle F\bar{z}^k, \bar{z}^k - z^\star\rangle$$
$$\text{(Assumption 3.2(iii))} \le 2\|z^k - z^\star\|^2 - \beta\gamma\big((1-\gamma^2L^2)\gamma + 2\rho - \beta\gamma\big)\|Fz^k\|^2$$
$$- \beta\gamma\big((1-\beta)\gamma + 2\rho\big)\|F\bar{z}^k\|^2 \tag{58}$$

To get a recursion it thus suffice to require

$$(1-\beta)\gamma + 2\rho > 0 \quad \text{and} \quad (1-\gamma^2L^2)\gamma + 2\rho - \beta\gamma \ge 0. \tag{59}$$

Rearranging and telescoping (58) achieves the claimed rate. Rearranging (59) completes the proof. $\square$

**Theorem 7.3.** *Suppose* $\mathrm{Alg}: \mathbb{R}^d \to \mathbb{R}^d$ *is quasi-nonexpansive. Then* $(z^k)_{k\in\mathbb{N}}$ *generated by* (LA) *converges to some* $z^\star \in \mathrm{fix}\,\mathrm{Alg}$.

*Proof.* By the composition rule (Bauschke & Combettes, 2017, Prop. 4.49(ii)) $\mathrm{Alg}^t$ is also nonexpansive. Since $(z^k)_{k\in\mathbb{N}}$ can be seen as a Krasnosel'skiĭ-Mann iteration of a quasi-nonexpansive operator the iterates converges to $z^\star \in \mathrm{fix}\,\mathrm{Alg}^t$ by Theorem C.1 with $\varepsilon_k = 0$, i.e. $\|z^k - z^\star\| \overset{k\to\infty}{\longrightarrow} 0$. By Bauschke & Combettes (2017, Prop. 4.49(i)) it also follows that $\mathrm{fix}\,\mathrm{Alg}^t = \mathrm{fix}\,\mathrm{Alg}$, which completes the proof. $\square$

**Corollary 7.5.** *Suppose* $F$ *is* $1/L$*-cocoercive. Then* $(z^k)_{k\in\mathbb{N}}$ *generated by* LA-GDA *with* $\gamma \le 2/L$ *converges to some* $z^\star \in \mathrm{zer}\,F$.

*Proof.* If $F$ is $1/L$-cocoercive then $\gamma F$ is $1/2$-cocoercive given $\gamma \le 2/L$, which in turn implies that $V = \mathrm{id} - \gamma F$ is nonexpansive. The claim follows from Theorem 7.3 and by observing that $\mathrm{fix}\,V = \mathrm{zer}\,F$. $\square$

Consider the forward-backward-forward (FBF) method of Tseng (1991). We can write one step as follows

$$\bar{z} = (\mathrm{id} + \gamma A)^{-1} Hz \tag{60a}$$
$$\mathrm{FBF}(z) = z - (Hz - H\bar{z}) \tag{60b}$$

where $H = \mathrm{id} - \gamma F$. The extragradient method is obtained as a special case when $A \equiv 0$.

**Theorem F.1.** *If* $A: \mathbb{R}^d \rightrightarrows \mathbb{R}^d$ *is maximally monotone and* $F: \mathbb{R}^d \to \mathbb{R}^d$ *is monotone and* $L$*-Lipschitz continuous then the operator* (60) *with* $\gamma \le 1/L$ *is quasi-nonexpansive. Furthermore,* $\mathrm{fix}\,\mathrm{FBF} = \mathrm{zer}\,S$ *with* $S := A + F$.

*Proof.* By $1/2$-cocoercivity from Lemma B.4 we obtain

$$\langle H\bar{z} - Hz, z - z^\star\rangle = \langle H\bar{z} - Hz, \bar{z} - z^\star\rangle + \langle H\bar{z} - Hz, z - \bar{z}\rangle$$
$$\text{(Lemma B.4)} \le \langle H\bar{z} - Hz, \bar{z} - z^\star\rangle - \tfrac{1}{2}\|H\bar{z} - Hz\|^2 - \tfrac{1}{2}(1-\gamma^2L^2)\|\bar{z} - z\|^2$$
$$\text{(monotone)} \le -\tfrac{1}{2}\|H\bar{z} - Hz\|^2 - \tfrac{1}{2}(1-\gamma^2L^2)\|\bar{z} - z\|^2 \tag{61}$$

The operator in (60b) satisfies

$$\|\mathrm{FBF}(z) - z^\star\|^2 = \|z - z^\star\|^2 + \|H\bar{z} - Hz\|^2 + 2\langle H\bar{z} - Hz, z - z^\star\rangle$$
$$\text{(61)} \le \|z - z^\star\|^2 - (1-\gamma^2L^2)\|\bar{z} - z\|^2$$

where the last term is negative due to $\gamma \le 1/L$. Recognizing the definition of quasi-nonexpansive completes the proof. $\square$

**Corollary 7.7.** *Suppose Assumption 3.2 holds. Then* $(z^k)_{k\in\mathbb{N}}$ *generated by* LA-CEG+ *with* $\lambda \in (0,1)$, $\gamma \in (\lfloor -2\rho\rfloor_+, 1/L)$ *and* $\alpha \in (0, 1 + \frac{2\rho}{\gamma})$ *converges to some* $z^\star \in \mathrm{zer}\,S$.

*Proof.* Quasi-nonexpansiveness of the operator $\text{CEG}^+ : \mathbb{R}^d \to \mathbb{R}^d$ follows from Theorem G.1(i) provided $\alpha \in (0, 1 + \frac{2\rho}{\gamma})$ so Theorem 7.3 applies.

It remains to verify that $\text{fix} \, \text{CEG}^+ = \text{zer} \, S$. This follows from

$$\tfrac{1}{\gamma}(Hz - H\bar{z}) \in A(\bar{z}) + F(\bar{z}) = S(\bar{z}), \tag{62}$$

and noticing that the stepsizes are positive, i.e. $\alpha > 0$ and $\gamma > 0$, which completes the proof. $\qquad\square$

# G  Analysis of CEG+

This section provides a simplified convergence proof of the CEG+ scheme proposed in Pethick et al. (2022, Cor. 3.2) without going through adaptivity and a projected interpretation. We additionally provide convergence in terms of the residual $\|z^k - \bar{z}^k\|$. The algorithm can be described with the following recursion

$$\bar{z}^k = (\text{id} + \gamma A)^{-1}(Hz^k) \tag{63a}$$

$$z^{k+1} = z^k - \alpha \left(Hz^k - H\bar{z}^k\right) \tag{63b}$$

where $H = \text{id} - \gamma F$. The EG+ algorithm is obtained as a special case when $A \equiv 0$.

**Theorem G.1.** *Suppose Assumption 3.2 and* $\gamma \in (\lfloor -2\rho \rfloor_+, 1/L]$. *Consider the sequence* $(z^k)_{k \in \mathbb{N}}$ *generated by* (63). *Then, for all* $z^\star \in \text{zer} \, S$, *it follows that*

*(i) the iterates* $(z^k)_{k \in \mathbb{N}}$ *satisfies*

$$\|z^{k+1} - z^\star\|^2 \leq \|z^k - z^\star\|^2 - \alpha(1 + \tfrac{2\rho}{\gamma} - \alpha)\|H\bar{z}^k - Hz^k\|^2 - \alpha(1 - \gamma^2 L^2)\|\bar{z}^k - z^k\|^2,$$

*and in particular,* $\text{CEG}^+ : \mathbb{R}^d \to \mathbb{R}^d$ *in* (12) *is quasi-nonexpansive if* $\alpha \in (0, 1 + \tfrac{2\rho}{\gamma})$.

*(ii) for* $\alpha \in (0, 1]$ *and* $\alpha < 1 + \tfrac{2\rho}{\gamma}$

$$\frac{1}{K} \sum_{k=0}^{K-1} \|z^k - \bar{z}^k\|^2 \leq \frac{\|z^0 - z^\star\|^2}{\alpha(1 - \gamma^2 L^2)K}. \tag{64}$$

*(iii) for* $\alpha \in (0, 1)$ *and* $\alpha < 1 + \tfrac{2\rho}{\gamma}$

$$\frac{1}{K} \sum_{k=0}^{K-1} \text{dist}(0, S\bar{z}^k)^2 \leq \frac{\|z^0 - z^\star\|^2}{\alpha\gamma^2(1 + \tfrac{2\rho}{\gamma} - \alpha)K}. \tag{65}$$

*Proof.* By $1/2$-cocoercivity of $H = \text{id} - \gamma F$ from Lemma B.4 we obtain

$$\langle H\bar{z}^k - Hz^k, z^k - z^\star \rangle = \langle H\bar{z}^k - Hz^k, \bar{z}^k - z^\star \rangle + \langle H\bar{z}^k - Hz^k, z^k - \bar{z}^k \rangle$$
$$\leq \langle H\bar{z}^k - Hz^k, \bar{z}^k - z^\star \rangle - \tfrac{1}{2}\|H\bar{z}^k - Hz^k\|^2 - \tfrac{1}{2}(1 - \gamma^2 L^2)\|\bar{z}^k - z^k\|^2 \tag{66}$$

The update in (63b) yields

$$\|z^{k+1} - z^\star\|^2 = \|z^k - z^\star\|^2 + \alpha^2\|H\bar{z}^k - Hz^k\|^2 + 2\alpha\langle H\bar{z}^k - Hz^k, z^k - z^\star \rangle$$
$$\overset{(66)}{\leq} \|z^k - z^\star\|^2 - 2\alpha\langle Hz^k - H\bar{z}^k, \bar{z}^k - z^\star \rangle$$
$$\quad - \alpha(1 - \alpha)\|H\bar{z}^k - Hz^k\|^2 - \alpha(1 - \gamma^2 L^2)\|\bar{z}^k - z^k\|^2. \tag{67}$$

Noticing that both latter terms are negative. Observe that by (63a) we have

$$\tfrac{1}{\gamma}(Hz^k - H\bar{z}^k) \in A(\bar{z}^k) + F(\bar{z}^k) = S(\bar{z}^k).$$

Therefore, by cohypomonotonicity of $S = A + F$,

$$\tfrac{1}{\gamma}\langle Hz^k - H\bar{z}^k, \bar{z}^k - z^\star \rangle \geq \rho\|Hz^k - H\bar{z}^k\|^2. \tag{68}$$

and consequently (67) leads to Fejér monotonicity,

$$\|z^{k+1} - z^\star\|^2 \leq \|z^k - z^\star\|^2 - \alpha(1 + \tfrac{2\rho}{\gamma} - \alpha)\|H\bar{z}^k - Hz^k\|^2 - \alpha(1 - \gamma^2 L^2)\|\bar{z}^k - z^k\|^2.$$

By telescoping we obtain the two claims. $\qquad\square$

# H  Experiments

## H.1  Simulations

We repeat the synthetic examples for convenience below.

**Example H.1** (PolarGame (Pethick et al., 2022, Ex. 3(iii))). *Consider*

$$Fz = (\psi(x, y) - y, \psi(y, x) + x),$$

*where* $\|z\|_\infty \leq {}^{11}/{}_{10}$ *and* $\psi(x, y) = \frac{1}{16}ax(-1 + x^2 + y^2)(-9 + 16x^2 + 16y^2)$ *with* $a = \frac{1}{3}$.

**Example H.2** (Quadratic (Pethick et al., 2022, Ex. 5)). *Consider,*

$$\min_{x \in \mathbb{R}} \max_{y \in \mathbb{R}} \phi(x, y) := axy + \frac{b}{2}x^2 - \frac{b}{2}y^2, \tag{69}$$

*where* $a \in \mathbb{R}_+$ *and* $b \in \mathbb{R}$.

The problem constants in Example H.2 can easily be computed as $\rho = \frac{b}{a^2+b^2}$ and $L = \sqrt{a^2 + b^2}$. We can rewrite Example H.2 in terms of $L$ and $\rho$ by choosing $a = \sqrt{L^2 - L^4\rho^2}$ and $b = L^2\rho$.

We provide below a slight generalization of the Forsaken example (Hsieh et al., 2021, Example 5.2), from which we derive another important case.

**Example H.3.** *Consider,*

$$\min_{|x| \leq {}^3/{}_2} \max_{|y| \leq {}^3/{}_2} \phi(x, y) := x(y - a) + \psi(x) - \psi(y), \tag{70}$$

*where* $\psi(z) = \frac{1}{4}z^2 - \frac{1}{2}z^4 + \frac{1}{6}z^6$ *and* $a \in \mathbb{R}$. *We have the following important cases:*

  *(i)  for* $a = 0.45$ *we recover Forsaken (Hsieh et al., 2021, Example 5.2).*

  *(ii)  for* $a = 0.34$ *we ensure that the first-order stationary point is a local Nash equilibrium (LNE), which is apparent from inspection of the Jacobian. We call this new example LNEForsaken.*

In both Example H.2 and Example H.3 the operator $F$ is defined as $Fz = (\nabla_x\phi(x, y), -\nabla_y\phi(x, y))$. For Example H.3 the Lookahead methods use $\tau = 20$, $\lambda = 0.2$ and $\gamma = {}^1/{}_L \approx 0.08$ and (R)APP uses $\tau = 10$, $\lambda = 0.2$ and $\gamma = {}^4/{}_L \approx 0.32$. In Examples H.1 and H.2 we use $\gamma = {}^1/{}_L$, $\lambda = 0.1$ for LA-GDA and EG+ with $\alpha = 0.1$ for the latter. In the constrained examples $L$ refers to the Lipschitz constant constrained to the constraint set.

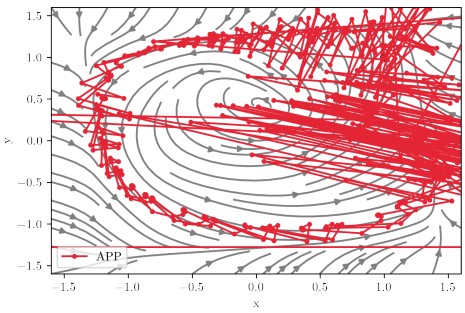

Figure 6: The iterates of APP associated with Figure 2.

## H.2  Image generation

**Architecture**  The ResNet uses a 128-dimensional input space for the generator and spectral normalization for the discriminator (see Chavdarova et al. (2020), Table 7)). The models' parameters are initialized using the Xavier initialization as suggested in Miyato et al. (2018).

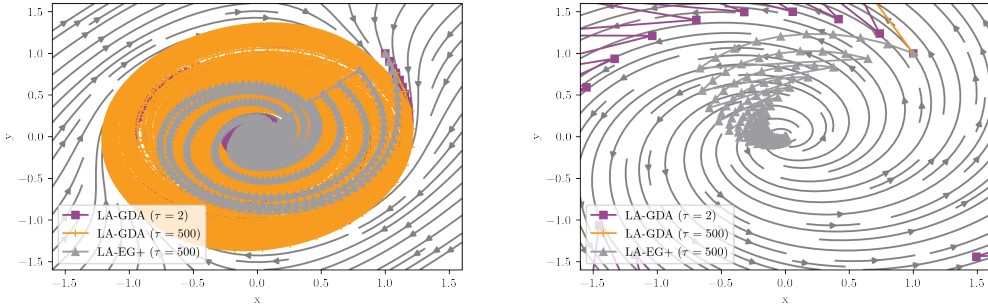

Figure 7: Iterates associated with Figure 3.

**Optimizers** All methods relies on stochastic gradients computed over a mini-batch. The discriminator and generator is updated in an alternating fashion. We use the same variant of extragradient as Chavdarova et al. (2020) uses in their implementation. The variant only uses the extrapolated point of the *opponent* in the update of the next iterate $(x^{k+1}, y^{k+1})$ as follows

$$
\begin{aligned}
\bar{x}^k &= x^k - \gamma_1 \nabla \phi(x^k, y^k) \\
\bar{y}^k &= y^k + \gamma_2 \nabla \phi(x^k, y^k) \\
x^{k+1} &= x^k - \gamma_1 \nabla \phi(x^k, \bar{y}^k) \\
y^{k+1} &= y^k + \gamma_2 \nabla \phi(\bar{x}^k, y^k)
\end{aligned}
\tag{71}
$$

Interestingly, we observed that the classical extragradient method (both a simultaneous and alternating variant) did not perform well under the hinge loss as used in the experiments. We leave investigate of this for future work.

**Evaluation** We use the Fréchet inception distance (FID) (Heusel et al., 2017) evaluated on $50\,000$ examples and the Inception score (ISC) (Salimans et al., 2016). For consistent and reproducible evaluation we use the `torch-fidelity` Python library (Obukhov et al., 2020) to compute the scores. The mean and standard deviation is computed over 5 and 3 independent execution in Table 2 and Table 3, respectively.

**Compute time** Producing Table 2 alone takes roughly 6 methods $\times$ 5 runs $\times$ 30 hours $= 37.5$ days on a NVIDIA A100 GPU.

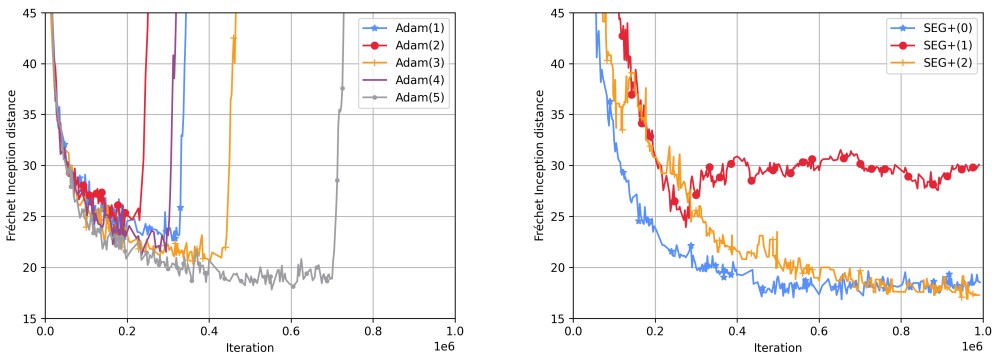

Figure 8: (left) Adam eventually diverges for all 5 runs. See Figure 4 for comparison with Lookahead. (right) In contrast, EG+ increases stability (and thus avoids divergence), but in effect might also be stuck in a local (suboptimal) solution. This explains the high variance and poor performance of EG+. By excluding the locally stuck run, EG+ achieves a FID of $16.88 \pm 0.05$ and a ISC of $8.0 \pm 0.02$, which is competitive even with the Lookahead-based methods.

### H.2.1 Hyperparameters

Table 4: Training Hyperparameters for Adam-based experiments on CIFAR10

| Hyperparameter | Adam | LA-Adam | ExtraAdam+ | LA-ExtraAdam+ | ExtraAdam | LA-ExtraAdam |
|:---:|:---:|:---:|:---:|:---:|:---:|:---:|
| lrD | 2e-4 | 2e-4 | 2e-4 | 2e-4 | 2e-4 | 2e-4 |
| lrG | 2e-4 | 2e-4 | 2e-4 | 2e-4 | 2e-4 | 2e-4 |
| Batch Size | 128 | 128 | 128 | 128 | 128 | 128 |
| $\beta_1$ | 0.0 | 0.0 | 0.0 | 0.0 | 0.0 | 0.0 |
| D-steps | 5 | 5 | 5 | 5 | 5 | 5 |
| Lookahead $\tau$ | | 5 | | 5000 | | 5000 |
| Lookahead $\lambda$ | | 0.5 | | 0.5 | | 0.5 |
| EG+ $\alpha$ | | | 0.5 | 0.5 | | |

Table 5: Training Hyperparameters for GDA-based experiments on CIFAR10

| Hyperparameter | GDA | LA-GDA | EG+ | LA-EG+ | EG | LA-EG | RAPP |
|:---:|:---:|:---:|:---:|:---:|:---:|:---:|:---:|
| lrD | 0.1 | 0.1 | 0.1 | 0.1 | 0.1 | 0.1 | 0.1 |
| lrG | 0.02 | 0.02 | 0.02 | 0.02 | 0.02 | 0.02 | 0.02 |
| Batch Size | 128 | 128 | 128 | 128 | 128 | 128 | 128 |
| D-steps | 1 | 1 | 1 | 1 | 1 | 1 | 1 |
| Lookahead $\tau$ | | 5000 | | 5000 | | 5000 | |
| Lookahead $\lambda$ | | 0.5 | | 0.5 | | 0.5 | |
| EG+ $\alpha$ | | | 0.5 | 0.5 | | | |
| RAPP $\tau$ | | | | | | | 3 |
| RAPP $\lambda$ | | | | | | | 0.9 |