# OpenReview forum: "Stable Nonconvex-Nonconcave Training via Linear Interpolation"
_NeurIPS.cc/2023/Conference — NeurIPS 2023 spotlight_

### Official Review · Reviewer_RmjV · 2023-06-29

**Soundness:** 3 good
**Presentation:** 2 fair
**Contribution:** 3 good
**Rating:** 6
**Confidence:** 2

**Summary:**

This paper applies linear interpolation to make neural networks stable. Based on the analysis of instabilities, the authors propose a new optimization scheme, RAPP. RAPP achieves last-iterate convergence rates for the full range of cohypomonotone problems. Moreover, by replacing the inner optimizer in RAPP, the authors propose a new Lookahead algorithm, which is helpful for expanding cohypomonotone problems. The authors also do experiments to verify that linear interpolation is beneficial to RAPP and Lookahead.

**Strengths:**

This paper applies linear interpolation to make neural networks stable. Based on the analysis of instabilities, the authors propose a new optimization scheme, RAPP. RAPP achieves last-iterate convergence rates for the full range of cohypomonotone problems. Moreover, by replacing the inner optimizer in RAPP, the authors propose a new Lookahead algorithm, which is helpful for expanding cohypomonotone problems. The authors also do experiments to verify that linear interpolation is beneficial to RAPP and Lookahead.

**Weaknesses:**

1. As illustrated in definition 4.1, formulation (4) is incorrect, because $z$ and $z^’$ have different dimensions.

2. In Theorem 5.1, the authors claim that Algorithm 1 converges to a neighborhood linearly, but, according to formulation (6), the upper bound of the neighborhood is $\infty$ when $\gamma\to 1/L$. This bound is too loose.

3. The authors didn’t mention linear interpolation in their theorems, but they add linear interpolation in their experiments. The theorems about linear interpolation are lacking.


4. The authors propose an optimization scheme RAPP and compare it to Adam. Then, the comparison with other optimization schemes lacked, such as Momentum, RMSprop, Adadelta, and so on.


**Questions:**

1. What’s the meaning of “id”?

**Limitations:**

See the weakness.

---

> ### Author Rebuttal · Authors · 2023-08-09
>
>
> We thank the reviewer for their valuable feedback and address all remaining concerns below:
>
> - **Linear convergence** when we have $\|w^{\tau+1}-w^\star\| \leq c \|w^\tau-w^\star\|$ for $c \in (0,1)$ it is standard to refer to it as linear convergence, but we will add a remark clarifying that the rate suffers if $\gamma$ is too close to $1/L$. Unfortunately this dependency is unavoidable and it also appears even for average iterate results of extragradient based methods (through e.g. $\kappa$ in Cor. 3.2 of [Pethick et al. 2021](https://openreview.net/pdf?id=2_vhkAMARk)).
> - **Missing theory for linear interpolation** We think there might be a misunderstanding. The methods RAPP, IKM, EG+ and Lookahead are all using linear interpolation and are analysed in Lemma 6.1, Theorem 6.2, Corollary 6.4 and the theorem statements in Section 7. So the theoretical results covers the linear interpolation used in the algorithms.
> - **Additional baseline** As suggested we additionally tried SGD with momentum. We started with the the default of 0.9, but ended up consecutively decreasing the momentum parameter due to instability. As can be inferred from the table below, the only stable run we obtained was for momentum=0.1, for which the FID was still worse than for SGD without momentum (see Table 3 in the paper).
>
>     |  momentum  | 0.9 | 0.5 | 0.1 |
>     |----| -------- | -------- | -------- |
>     |FID | 319.52   | 153.39     | 20.90  |
>
>     Note that in the paper we already include comparison with both Adam and SGD.
>
>
> Minor comments:
>
> - **Typo in equation (4)** That is indeed a typo – the $n$ should have been $d$ (the correct definition can be found in Definition B.1 of the appendix). Thanks for catching it.
> - **Meaning of “id”** It refers to the identity mapping. We will add the definition to Appendix B where the rest of the definitions can be found.

---

> > ### Comment · Reviewer_RmjV · 2023-08-17
> >
> > Thanks for your reply. To be honest, your paper is difficult to follow. Other reviewers also lack confidence in your paper due to their lower confidence rate. Moreover, according to the definition of linear convergence rate in Wikipedia, your answers are unsatisfactory. When $\gamma\to 1/L$, the second term of (6) becomes $\infty$. And the definition of linear convergence rate doesn't include the second term. In addition, the baselines are still few. In summary, I keep my rating.

---

> > > ### Author Response · Authors · 2023-08-18
> > >
> > > Thank you for following up on the rebuttal!
> > >
> > > It is true that the _size of the neighborhood_ that we converge to linearly increases with $\gamma$, but in the deterministic case $\sigma$ is zero so we converge exactly. In the stochastic case (treated in the appendix) we control the term through $\sigma$.
> > >
> > > Concerning the baselines we agree that more comparisons never hurts, but we had to draw the line somewhere due to computational constraints. Each run takes 30 hours and we average over 5 independent runs for each configuration due to high variance common to GAN training. With that said, we do compare with 4 strong baselines: GDA, EG, ExtraAdam and Adam (which amounts to almost a month of compute just for the baselines).

---

### Official Review · Reviewer_KxAB · 2023-07-03

**Soundness:** 4 excellent
**Presentation:** 4 excellent
**Contribution:** 4 excellent
**Rating:** 6
**Confidence:** 3

**Summary:**

The paper gives a theoretical analysis of linear interpolation that can help stabilize neural network training. They show these instabilities in the optimization are caused by nonmonotonicity in the loss landscape. They also construct a new optimization scheme, called "relaxed approximate proximal point" (RAPP) which is the first explicit method to obtain last iterate convergence rates for cohypomonotone problems. They also show that the extragradient+ algorith, RAPP, and the Lookahead based algorithms are all instances of KM iteration. They show experiments on synthetic problems and GANs.

**Strengths:**

- Gives a nice summary of the problem setting and also clearly states what the paper is trying to resolve.

- Resolves questions that were implied by the literature, and thus would be interesting to other researchers.

- Gave nice proofs that generalize a number of methods as instances of the (inexact) KM iteration.

**Weaknesses:**

- An analysis of the wallclock time for the algorithms would have been nice.

- It would have been nice to see experiments with different hyperparameters for the Adam optimizer, such as different learning rates or $\beta$ parameters.

- In the abstract: I'm not sure I am convinced that "linear interpolation as a key method for stabilizing (large-scale) neural network training." The experiments haven't convinced me of this. I would probably rephrase it as "helping stabilize" or something similar.


**Questions:**

- How did the wallclock time of Adam compare with the various Lookahead and Extragradient algorithms?

- I know the authors mention that the point of the GAN experiment is not to beat the state-of-the-art, but are there any implications of these algorithms for achieving state-of-the-art performance?

**Limitations:**

- The authors don't really explicitly state their limitations, except perhaps to not compare their GAN experiment to the state-of-the-art.

---

> ### Author Rebuttal · Authors · 2023-08-09
>
> We thank the reviewer for their valuable feedback and address all remaining concerns below:
>
> - **Wallclock time** The wallclock time is essentially made to be the same across all methods in the experiments by providing each method with the same number of gradient computations (see line 263-267). All the methods (Adam, SGD, Lookahead, extragradient methods and RAPP) carry out the same operator repeatedly: addition of a gradient and an iterate. The only (crucial) difference is *what* gradient and iterate are involved. So all the methods are provided with the same computational budget.
> - **Adam hyperparameters** We note that we have used the *optimized* hyperparameters of Adam from Chavdarova et al. (2020) (see [below Table 10](https://arxiv.org/pdf/2006.14567.pdf)).
> - **Implication for state-of-the-art** It is definitely interesting to try RAPP at larger scale, especially in transformer-based setups where tricks like gradient clipping are otherwise used for preventing divergence (see e.g. [StyleSwin](https://openaccess.thecvf.com/content/CVPR2022/papers/Zhang_StyleSwin_Transformer-Based_GAN_for_High-Resolution_Image_Generation_CVPR_2022_paper.pdf) and the associated code).
> - **Abstract** We will rephrased the abstract as suggested

---

> > ### Comment · Reviewer_KxAB · 2023-08-15
> >
> > Thanks for your reply. It definitely cleared up some questions. I'll keep my rating.

---

### Official Review · Reviewer_TKoG · 2023-07-04

**Soundness:** 3 good
**Presentation:** 3 good
**Contribution:** 3 good
**Rating:** 7
**Confidence:** 1

**Summary:**

This paper studies the global convergence problem under cohypomonotonicity structural assumption. The authors prove the global convergence rate for the last iterate of their proposed algorithm RAPP. RAPP is the first explicit scheme to
58 have non-asymptotic guarantees for the full range of cohypomonotone problems.

**Strengths:**

1. This paper proves global convergence rates for the last iterate of our proposed algorithm RAPP which addi57 tionally handles constrained and regularized settings.
2. By replacing the inner optimization routine in RAPP with GDA and EG, this paper rediscovers the Lookahead algorithms. Guarantees for the Lookahead variants by deriving nonexpansive properties of the base optimizers are also obtained.
3. Their algorithm is tested experimentally.

**Weaknesses:**

1. They have only dealt with compositions of operators.
2. Whether RAPP could be extended to more general cases is not clear.


**Questions:**

N/A

**Limitations:**

Listed in Weakness section.

---

> ### Author Rebuttal · Authors · 2023-08-09
>
> We thank the reviewer for their valuable feedback.
> We agree that it is definitely interesting to see if we can extend the results further and we indeed have some preliminary positive results going beyond compositions.

---

> > ### Comment · Reviewer_TKoG · 2023-08-14
> >
> > Thanks for the reply. I will keep my rating.

---

### Official Review · Reviewer_7YGi · 2023-07-06

**Soundness:** 4 excellent
**Presentation:** 2 fair
**Contribution:** 3 good
**Rating:** 7
**Confidence:** 3

**Summary:**

This paper continues a line of work motivated by the need to design algorithms for non-convex non-concave min-max problems. Such problems arise in the training of GANs as well as reinforcement learning via self-play. Since solving general non-convex non-concave min-max problems is intractable, the main approach in this line of work is to study the subclass of problems for which the gradient map (in the constrained case one additionally adds a map corresponding to the normal vectors of the constraints) is a cohypomonotone operator. Even for problems in this subclass previously developed algorithms can be shown to diverge.

The main contribution of this paper is a unified way of designing algorithms for finding zeros of cohhypomonotone operators in terms of the KM iteration from the theory of non-monotone operators. This leads to several new results including (1) a new algorithm RAPP which handles constraints and has a convergence rate independent of the cohypomonotonicity parameter $\rho$, and (2) a simple analysis in terms of the KM iteration for the Lookahead algorithm, which additionally provides theoretical justification for the empirically most useful setting of the main hyperparameter for Lookahead.

**Strengths:**

1. The paper provides both a new algorithm that provably converges for a broader class of problems (i.e. those with constraints and no dependence on the cohypomontonicity parameter) than previously known.
2. The paper gives a simple and intuitive general framework for analyzing optimization algorithms for cohypomonotone operators, which gives additional insight into previously discovered algorithms (in particular Lookahead).
3. There are also experimental results illustrating the convergence advantage of such algorithms, including in the setting of large-scale GAN training setting.

**Weaknesses:**

The paper is a little bit short on exposition explaining why finding the zeros of cohypomontone operators is important. Of course this strictly generalizes solving convex-concave min-max problems, but exactly how much it generalizes is not stated.

In particular, Example 3.1 gives an example of how the operator arises from a min-max problem, but does not explain when/why this operator will satisfy the conditions of Assumption 3.2. While the empirical results in the paper justify that there are practical min-max where the proposed algorithms converge while other approaches diverge, it would be nice to have a simple analytic example to illustrate more clearly what types of min-max problems give rise to cohypomonotone operators.

**Questions:**

1. Is there a simple example of a non-concave-non-convex min-max problem so that it is easy to compute by hand that the associated operator is cohypomonotone?
2. Is there a (even hand-wavy) theoretical reason to expect that cohypomonotonicity is a good fit for modelling large-scale min-max problems arising for GANs?

**Limitations:**

Yes.

---

> ### Author Rebuttal · Authors · 2023-08-09
>
> We thank the reviewer for their valuable feedback and address all remaining concerns below:
>
> **Generality of cohypomonotonicity and simple examples** One of the simplest examples is probably Example H.2 in the appendix where we also provide the closed-form solution to $\rho$ and $L$. Another simple condition is the one proposed in [Example 1, Lee & Kim 2021](https://arxiv.org/pdf/2106.02326.pdf) which shows that a certain problem class is cohypomonotone when a second order condition is satisfied (known as the interaction dominant condition).
>
> Additionally, all the results extend to weak MVIs apart from the last iterate result of RAPP (see Appendix B.1 where we discuss this). So other examples includes the two-agent zero-sum reinforcement learning problem mentioned in [Diakonikolas et al. 2021](https://arxiv.org/pdf/2011.00364.pdf), all quasi-convex-concave and all star-convex-concave problems. Note that for the last iterate result we only assume that cohypomonotonicity holds for all pairs of points for simplicity. We really only need the condition to hold for each *consecutive pair of points* of the iterates, so we could expect this to hold even in many weak MVIs.
>
> **Intuition behind applying to GANs** Our analysis primarily relies on application of cohypomonotonicity between a given iterate $z^k$ and some solution $z^\star$. Let us for simplicity consider the unconstrained case where the condition reduces to
>
> $\langle Fz^k, z^k-z^\star \rangle \geq \rho\|Fz^k\|^2$
>
> This allows the gradients to point away from the solution set (when $\rho$ is negative), which is generally the case in nonconvex landscapes. The slack provided on the angle is particularly large when the norm of the gradient is large, which is usually the case in the beginning of training. In general the slack can be larger when the stepsize is large through the relationship $\rho > -2/\gamma$ (large stepsizes of e.g. 0.1 is what is used in practice). It is definitely interesting to see if the condition can be relaxed further.
>
> Another motivation for us was the instability at the solutions illustrated in [Figure 5, Berard 2020](https://arxiv.org/pdf/1906.04848.pdf) by inspection of the Hessian/Jacobian. Linear interpolation can be seen as a way to stabilize the dynamics also locally. We will comment on this in the final version.

---

> > ### Comment · Reviewer_7YGi · 2023-08-16
> >
> > Thanks for your response clarifying the examples of cohypomonotonicity and the application to GANs. I will keep my score.

---

### Decision · Program_Chairs · 2023-09-21

**Decision:**

Accept (spotlight)

**Comment:**

All reviewers suggested acceptance. Moreover, I agree with the evaluations. There is no reason to be verbose in this metareview; I can simply just point to the original reviews. Congratulations on a nice paper.